# Progressive Mitochondrial SOD1^G93A^ Accumulation Causes Severe Structural, Metabolic and Functional Aberrations through OPA1 Down-Regulation in a Mouse Model of Amyotrophic Lateral Sclerosis

**DOI:** 10.3390/ijms22158194

**Published:** 2021-07-30

**Authors:** Iago Méndez-López, Francisco J. Sancho-Bielsa, Tobias Engel, Antonio G. García, Juan Fernando Padín

**Affiliations:** 1Instituto Teófilo Hernando and Departamento de Farmacología y Terapéutica, Facultad de Medicina, Universidad Autónoma de Madrid (UAM), 28029 Madrid, Spain; iagomendez@hotmail.es (I.M.-L.); antonio.garcia@ifth.es (A.G.G.); 2Departamento de Ciencias Médicas, Facultad de Medicina, Universidad de Castilla-La Mancha (UCLM), 13005 Ciudad Real, Spain; francisco.sancho@uclm.es; 3Department of Physiology & Medical Physics, RCSI University of Medicine and Health Sciences, D02 YN77 Dublin, Ireland; tengel@rcsi.ie; 4FutureNeuro SFI Research Centre for Chronic and Rare Neurological Diseases, RCSI University of Medicine and Health Sciences, D02 YN77 Dublin, Ireland

**Keywords:** amyotrophic lateral sclerosis, SOD1^G93A^, chromaffin cell, mitochondrial dysfunction, OPA1

## Abstract

In recent years, the “non-autonomous motor neuron death” hypothesis has become more consolidated behind amyotrophic lateral sclerosis (ALS). It postulates that cells other than motor neurons participate in the pathology. In fact, the involvement of the autonomic nervous system is fundamental since patients die of sudden death when they become unable to compensate for cardiorespiratory arrest. Mitochondria are thought to play a fundamental role in the physiopathology of ALS, as they are compromised in multiple ALS models in different cell types, and it also occurs in other neurodegenerative diseases. Our study aimed to uncover mitochondrial alterations in the sympathoadrenal system of a mouse model of ALS, from a structural, bioenergetic and functional perspective during disease instauration. We studied the adrenal chromaffin cell from mutant SOD1^G93A^ mouse at pre-symptomatic and symptomatic stages. The mitochondrial accumulation of the mutated SOD1^G93A^ protein and the down-regulation of optic atrophy protein-1 (OPA1) provoke mitochondrial ultrastructure alterations prior to the onset of clinical symptoms. These changes affect mitochondrial fusion dynamics, triggering mitochondrial maturation impairment and cristae swelling, with increased size of cristae junctions. The functional consequences are a loss of mitochondrial membrane potential and changes in the bioenergetics profile, with reduced maximal respiration and spare respiratory capacity of mitochondria, as well as enhanced production of reactive oxygen species. This study identifies mitochondrial dynamics regulator OPA1 as an interesting therapeutic target in ALS. Additionally, our findings in the adrenal medulla gland from presymptomatic stages highlight the relevance of sympathetic impairment in this disease. Specifically, we show new SOD1^G93A^ toxicity pathways affecting cellular energy metabolism in non-motor neurons, which offer a possible link between cell specific metabolic phenotype and the progression of ALS.

## 1. Introduction

Jean-Martin Charcot first described amyotrophic lateral sclerosis (ALS) as a syndrome in 1869, a disease characterized by motor neuron degeneration within the spinal cord, brainstem and motor cortex [1]. The demise of these neurons leads to muscle weakness, progressing to muscle paralysis and atrophy, and the patient ultimately dies due to respiratory failure usually within three to five years of diagnosis. The onset of ALS may occur either sporadically (as in 90% of cases) or due to different gene mutations that follow familial transmission (fALS). Although there are many genes known to be involved in fALS [2], the first mutation discovered produces a substitution of the glycine at position 93 by alanine (G93A) in Cu^2+^/Zn^2+^ superoxide dismutase (SOD1^G93A^). The identification of this mutation facilitated the development of the first transgenic mouse model of fALS [3], in which pathological and clinical symptoms similar to those seen in ALS patients develop, becoming the most studied mouse model of this disease. To date, at least 185 mutations in the SOD1 gene have been associated with ALS, responsible for approximately 2% of all cases (http://www.omim.org/entry/105400, access on 1 February 2021).

Early postmortem studies of the spinal cord of patients with ALS reveal the presence of vacuoles in both axons and dendrites; these derived from degenerating mitochondria [4]. Similar vacuoles were also detected in SOD1^G93A^ mutant mice [5,6,7,8], indicating that mitochondrial degeneration is an important early event triggering the decline of motor neurons and inducing cell death [7]. Mitochondria are the source of cellular energy, and they are involved in the generation of reactive oxygen species (ROS), in intracellular calcium homeostasis, calcium-mediated excitotoxicity and the intrinsic apoptotic pathway. All these mitochondrial functions are affected in ALS, which suggest mitochondria are a convergence point in the disease. Thus, ALS is currently considered a human secondary mitochondriopathy [9].

ALS was traditionally thought to be exclusively a motor neuron pathology, yet accumulating evidence from the SOD1^G93A^ transgenic mouse model indicates that the demise of motor neurons derives from damage to other cell types in the central nervous system (CNS), such as microglia, macrophages or oligodendrocytes. Thus, the concept of “*non-autonomous motor neuron death*” was recently devised to explain the need for additional cell types to be affected in order to produce motor neuron death [10]. Indeed, like SOD1, other proteins altered in fALS are expressed ubiquitously, such as the vesicle-associated membrane protein-associated protein B (VAPB), the transactive response (TAR)-DNA binding protein 43 kDa (TDP-43) or profilin 1 (PFN1), and in consequence suggest for all of them a ubiquitous alteration. Thereby, in the SOD1^G93A^ mice, the expression of this mutated protein was accumulated beyond the motor neurons, as in non-nervous tissue such as the liver [11] and skeletal muscle [12]. However, while the contribution of those tissues to motor neuron degeneration and disease progression was initially unclear, compelling evidence has since then emerged that ALS is a “multisystemic” disease whereby structural, physiological and metabolic changes in different peripheral cell types or tissues may act mutually and synergistically to provoke disease onset and to define its severity [13,14,15,16,17]. Indeed, in early experiments the disease did not develop when the mutant SOD1 was expressed exclusively in either motor neurons or astrocytes [18,19], suggesting that the pathogenesis of fALS requires an interaction between motor neurons and their surrounding non-neuronal cells. Moreover, in patients and in SOD1^G93A^ mice, denervation of the muscle occurs prior to motor neuron degeneration, beginning at the distal axon and proceeding in a retrograde “*dying back*” pattern [20], as opposed to the “*dying-forward*” hypothesis [21]. Since then, several studies have provided evidence of the participation of peripheral tissues in ALS disease onset and progression. For instance, microRNAs released from peripheral tissues are thought to be involved in either inducing damage or repair at muscle junctions and in the spinal cord, depending on the tissue’s demands (reviewed in [12,22]).

Evidence for the implication of the autonomic nervous system (ANS) in ALS has become overwhelming, and neuromuscular junction alterations and cardiac sympathetic denervation appear to be initial events in ALS patients, prior to the appearance of motor symptoms [23,24]. Since these initial reports, there have been many descriptions of sub-clinical dysautonomia of cardiovascular, gastrointestinal, salivary, skin and lachrymal regulation in patients, even at early stages of the disease [25,26]. Such alterations were corroborated in vivo in SOD1^G93A^ transgenic mice [27], and furthermore, increased plasmatic norepinephrine is observed in both patients and mouse models [28,29]. However, enhanced catecholamine secretion is difficult to explain due to the preganglionic sympathetic denervation that occurs in the adrenal medulla [29] and in other organs [24,30]. As peripheral catecholamine levels are directly correlated with their secretion from chromaffin cells (CCs) in the adrenal medulla, alterations to these cells should be studied and could provide clues regarding the impaired CNS neurotransmission. At symptomatic stages in the SOD1^G93A^ transgenic mice, we detected slower exocytotic fusion pore opening, expansion and closure, with a higher quantal size of single exocytotic events in CCs stimulated with acetylcholine [31], which could explain the aforementioned findings. Alterations in the ANS, and therefore in the CCs, should not be underestimated since patients die as a result of sudden death syndrome because they are unable to compensate for cardiorespiratory arrest [32].

As SOD1 is a ubiquitous enzyme, we wanted to see if SOD1^G93A^ is accumulated in mitochondria of CCs. In addition, to define the mechanisms that might cause the disease, we studied the cells before (P30—postnatal day 30) and after (P120) symptoms of ALS development. As the mitochondria of CCs have not been studied previously, we focused on broadly describing the mechanisms that may be involved. As a first premise, we wanted to demonstrate the presence of the mutant SOD1^G93A^ protein in different compartments of the mitochondria, and once confirmed, we studied how the SOD1^G93A^ protein affects mitochondrial ultrastructure and function, assessing their metabolism, mitochondrial membrane potential (_m_ψ), fusion and the generation of free radicals. The study of ALS in those cells with a metabolism distinct to that of motor neurons, such as CCs in which glycolytic and oxidative metabolism are also important, may help us to better understand the metabolic impairment that occurs in this mitochondriopathy.

## 2. Results

### 2.1. Motor Disability and Weight Loss at P120 Supports the Symptomatic Status of the SOD1^G93A^ Mouse Model of ALS

The body weight and motor coordination of P30 and P120 mice were analyzed to validate the criteria used here to define the presymptomatic and symptomatic stages (Appendix A). Motor coordination was evaluated with the rotarod test, and significant differences in the latency to fall were only observed between the WT and transgenic mice at P120, the SOD1^G93A^ mice remaining on the rotating rod only half the time of the WT mice (45 ± 14 vs. 91 ± 5 s in WT). By contrast, there was no significant difference in the latency to fall of the P30 mutant and WT mice in this test (86 ± 6 vs. 77 ± 5 s in WT). Likewise, the body weight of the mice only differed significantly in the P120 group, the mean weight of which was nearly 25% less than that of the WTs (27.2 ± 0.3 g in WT vs. 21.0 ± 0.2 g in SOD1^G93A^).

### 2.2. Morphological Analysis of Mitochondria

The ultrastructure of mitochondria in the adrenal medulla of WT and SOD1^G93A^ mice was analyzed by TEM at both P30 and P120. We acquired images of whole CCs at 5000× magnification in which mitochondria were easily identified and their size could be quantified. To study mitochondrial ultrastructure, we acquired 40,000× higher resolution images that allowed us to distinguish all the mitochondrial membranes, including the cristae.

#### 2.2.1. Morphological Mitochondrial Damage in Chromaffin Cells from SOD1^G93A^ Mice

Mitochondria can adopt diverse shapes in the CCs of a P30 WT mouse: circular, elliptical or even branched (Figure 1A(a–d)). The internal ultrastructure of the mitochondria is predominantly found in an orthodox state (67.8 ± 2.2% of the mitochondrial population), characterized by an expanded matrix and typically revealing fairly regular spacing of the lamellar cristae, with a separation of 30.8 ± 2.8 nm. More condensed forms of mitochondria can co-exist (32.1 ± 2.2% of mitochondrial population), with a compact matrix and larger internal compartments connecting them or with a matrix extending throughout the intermembrane space. The interlamellar distance of condensed mitochondria is less than that in the orthodox state (9.9 ± 1.1 nm). Moreover, there are often multiple internal structures within a single mitochondrion, where the cristae tend to form tubes and there are short flat or larger lamellae.

Some differences were detected between P30 and P120 WT mice that may be related to aging. Specifically, mitochondrial swelling or vacuolization were found in 14.5 ± 0.6% of the mitochondrial population of P120 CCs. It is worth noting that vacuolization or swelling did not happen in the whole mitochondrial population in a homogeneous way, but it mainly occurred in those closely located to dilated rough endoplasmic reticulum (RER). Additionally, although rare, we found mitochondria with outer membrane leakage similar to those seen in SOD1^G93A^ at pre-symptomatic stages. However, aging did not appear to produce changes in the proportion of mitochondria adopting each structure, orthodox (67.7 ± 2.4%) or condensed (32.3 ± 2.4%) (the typical mitochondrial structures at P30 and P120 are shown in Figure 1A).

Inner mitochondrial matrix vacuolization and cristae swelling was more common in mitochondria from P30 SOD1^G93A^ mice (82.7 ± 3.4%) than in the WT P30 mice, which mostly contained orthodox mitochondria (86.0 ± 3.0%). Of the altered mitochondria, herniation occurred in 17.6 ± 0.2%, evident as an asymmetric bleb of the expanded mitochondrial membrane. It was notable that mitochondrial sprouting developed from the boundary of the intermembrane space in 16.3 ± 0.3% of cases (Figure 1B(d)), from the matrix in 15.5 ± 4.4% of cases (Figure 1B(b)) and from the mitochondrial cristae in a 68.1 ± 4.1% of cases (Figure 1B(c)). In only a small proportion of those blebs (4.21 ± 0.8%) were the outer mitochondrial membrane ruptured (Figure 1B(b,c)). Finally, approximately 8% of mitochondria were disorganized, with a characteristic disruption of the cristae (results from 318 images of mitochondria from at least four different animals). We did not observe notable differences in the internal ultrastructure in mitochondria from P30 and P120 SOD1^G93A^ mice. As at P30, the orthodox state was frequent at P120, with a greater proportion than the WT at the same age (*** p* < 0.01), with 88.4 ± 5.8% of mitochondria adopting this profile, although 21.6% of them had a disorganized ultrastructure and disrupted mitochondrial cristae. This was a similar proportion of mitochondrial cristae swelling to that at P30 (80.7 ± 6.5%), although there was a higher proportion of mitochondria with a broken outer mitochondrial membrane and the loss of content (7.9 ± 1.1 vs. 4.2 ± 0.8% of the WT; ** p* < 0.05). There was no difference in the sprouting of the mitochondrial matrix, intermembrane boundary spaces or cristae at P120 compared with that at P30. Moreover, there did not seem to be a relationship between outer mitochondrial membrane rupture and the swelling of the mitochondrial cristae, as those mitochondria with broken membranes were not herniated or extensively vacuolated.

#### 2.2.2. Lack of Mitochondrial Maturation in Chromaffin Cells of SOD1^G93A^ Mice Is Reflected by the Smaller Mitochondria Size Starting at a Presymptomatic Stage

The changes in mitochondrial size in CCs with age were analyzed in 5000× TEM images, in which whole CCs can be seen in the tissue, separated by their plasma membranes (Figure 2A–D). The number of mitochondria in the adrenal medulla tissue decreased with age in WT mice, from 0.370 ± 0.003 at P30 to 0.237 ± 0.002 mitochondria/μm^2^ of cytosol at P120 (** p* < 0.05; Figure 2E), although the average size of the mitochondria tended to increase from 0.218 ± 0.008 to 0.244 ± 0.014 μm^2^ (Figure 2F). This trend contrasted with the increase in mitochondrial density in SOD1^G93A^ CCs with age, which was significantly lower than in the WTs at P30 (0.299 ± 0.003 mitochondria/μm^2^ of cytosol, * *p* < 0.05) but higher at P120 (0.331 ± 0.019 mitochondria/μm^2^ of cytosol, ** *p* < 0.01). Moreover, mitochondria were always smaller, up to 15% smaller than in the WTs at presymptomatic stages and up to 32% lower at symptomatic stages (0.184 ± 0.009 μm^2^ at P30 and 0.165 ± 0.007 μm^2^ at P120, * *p* < 0.05 and ** *p* < 0.01, respectively). This reduced mitochondrial size and density during cell maturation was correlated with disease consolidation, to some extent reflecting an alteration in the fusion of the small mitochondria in SOD1^G93A^ CCs. Note the internal alterations to the mitochondria from SOD1^G93A^ mice compared with the WT mitochondria at both ages (Figure 2A–D), best appreciated in the magnified areas of high mitochondrial density (Figure 2A’–D’).

#### 2.2.3. Cristae Swelling and Increased Cristae Junctions Size in SOD1^G93A^ Chromaffin Cells

The ultrastructure of mitochondria was studied in high-resolution 40,000× TEM images in which individual mitochondria can be seen at both stages and obvious mitochondrial swelling was evident in the SOD1^G93A^ CCs (Figure 3A). While the number of cristae per µm^2^ of mitochondrial surface differed at the presymptomatic stage (44 ± 4 in WT vs. 34 ± 2 in SOD1^G93A^ at P30; * *p* < 0.05), no changes could be observed at symptomatic stages (51 ± 4 in WT vs. 60 ± 4 in SOD1^G93A^ at P120, Figure 3B). However, this was not the case for the cristae area (Figure 3C), which increased notably in the SOD1^G93A^ mitochondria relative to the WTs at P30 (0.0040 ± 0.0001 µm^2^ vs. 0.0070 ± 0.0005 µm^2^, *** *p* < 0.001) and P120 (0.0035 ± 0.0001 µm^2^ vs. 0.0077 ± 0.0006 µm^2^, *** *p* < 0.001). This means that the cristae area in SOD1^G93A^ mice was almost twice the physiological area, even at presymptomatic stages. Interestingly, the cristae area was constant at both ages in WT mice. To compare the cristae area relative to mitochondrial size, we calculated the proportion of mitochondria occupied by cristae (Figure 3D), which was 20% in WT CCs at both P30 and P120. This value rose significantly in SOD1^G93A^ CCs due to the swelling of the cristae, reaching 31% at presymptomatic P30 (** *p* < 0.01) and 47% at the symptomatic P120 stage (*** *p* < 0.001), a significant deterioration as the disease became established.

Another important parameter analyzed and described to be involved in mitochondrial architecture was the size of the cristae junction, due to its importance in cristae formation and mitochondrial function. We analyzed the mitochondria cristae to evaluate the cases in which the cristae junction was observed, and we measured the diameter of the pore that produces the inner mitochondria membrane invagination (Figure 4). The mean cristae junction in mitochondria of the CCs from WT mice was 13.6 ± 1.1 nm at P30 and 14.9 ± 1.3 nm at P120. This value increased by 72% in the presymptomatic mutated mice at P30 (23.7 ± 3.5 nm, ** *p* < 0.01) and by 206% in the symptomatic P120 mice (45.9 ± 10 nm, *** *p* < 0.001). Hence, the changes in this morphological parameter paralleled the disease-associated cristae swelling in the SOD1^G93A^ mice.

### 2.3. Functional Analysis of Mitochondria

Mitochondrial function was evaluated with specific tests related to their physiological function. We assessed the _m_ψ, redox balance and the bioenergetic status of the CCs. Together, these parameters gave us a general idea of the mitochondrial function in the SOD1^G93A^ CCs, both at presymptomatic and symptomatic stages.

#### 2.3.1. The Mitochondrial Membrane Potential Is Lower in CCs from SOD1^G93A^ Mice Starting at Presymptomatic Stages

The _m_ψ was measured using the cationic fluorescent dye TMRE, which accumulates in the mitochondria due to the organelle’s electronegativity and its free diffusion through biological membranes. During mitochondrial depolarization with the protonophore FCCP, we can monitor how the probe spreads from the mitochondria to the cytosol and thereby can calculate the changes in _m_ψ (Figure 5). After incubating CCs with the dye, the TMRE fluorescence highlights the mitochondrial electronegativity in the basal state, with more intense fluorescence reflecting a more electronegative state of the mitochondria. In 3D representations of individual CCs, peaks of fluorescence could be observed that corresponded to areas rich in mitochondria (Figure 5A), and these peaks were lower in SOD1^G93A^ mice, reflecting a smaller _m_ψ under basal conditions. When we calculated the mean basal fluorescence, it was 21.9% lower at P30 and 17.1% lower at P120 (*** *p* < 0.001 vs. WT for each of them). In both cases, we detected a dissipation of the _m_ψ during the FCCP pulse and the disappearance of the ensuing peaks. We quantified the change in the fluorescence registered during the protonophore pulse, which was smaller in the SOD1^G93A^ group relative to the WTs: 42.6% at P30 (20.3 ± 1.5 in SOD1^G93A^ vs. 35.3 ± 3.1 AFU in WT, * *p* < 0.05; Figure 5D) and 48.8% at P120 (26.2 ± 3.2 in SOD1^G93A^ vs. 51.2 ± 5.6 AFU in WT, *** *p* < 0.001; Figure 5E).

#### 2.3.2. Oxidative Cellular Stress in Chromaffin Cells from SOD1^G93A^ Mice Is Initiated during the Presymptomatic Stages and Augments as the Disease Progresses

The global cellular redox balance was studied in CCs using the CM-H_2_DCFDA fluorescent probe, the fluorescence signal of which is enhanced upon oxidation. CCs were incubated with the dye and the fluorescence was monitored in individual cells through an epifluorescence microscope (Figure 6). In this case, we acquired images over 30 min (t_30_, 1 min intervals) to evaluate the fluorescence changes due to basal cellular activity. The 3D reconstruction in an individual P30 (Figure 6A) and P120 cell (Figure 6B) at t_0_ and t_30_ clearly showed the increase in fluorescence after the 30 min protocol and the increased fluorescence in SOD1^G93A^ cells at both ages relative to those from WT CCs. We quantified data from more than 20 cells in at least four different cultures, comparing the increase in the mean fluorescence intensity from t_0_ to t_30_. We observed a higher rate of CM-H_2_DCFDA oxidation in SOD1^G93A^ CCs at both ages (26.7 ± 4.4 in WT vs. 43.2 ± 6.6 AFU in SOD1^G93A^ at P30, * *p* < 0.05; and 51.3 ± 5 in WT vs. 76.8 ± 5.8 AFU in SOD1^G93A^ at P120, ** *p* < 0.01: Figure 6E,G). When we compared the slope of the increase in cell fluorescence to calculate the rate of probe oxidation (Figure 6F,H), oxidation occurred faster with aging in the WT cells (0.76 ± 0.12 at P30 vs. 1.71 ± 0.17 AFU/min at P120), yet in the SOD1^G93A^, oxidation was always faster than in the WT cells of the same age (1.33 ± 0.22 AFU/min at P30, * *p* < 0. 05; and 2.67 ± 0.19 AFU/min at P120, *** *p* < 0.001). This means that the rate of oxidation in SOD1^G93A^ CCs doubles as the disease is consolidated.

#### 2.3.3. Cellular Bioenergetics Is Impaired in SOD1^G93A^ Chromaffin Cells

To study the bioenergetic profiles in our model system and their evolution with age and pathology, we performed a mitochondrial stress test by measuring the oxygen consumption rate (OCR) in a Seahorse XFp analyzer. With different drugs targeting specific mitochondrial functions, we analyzed the changes in OCR, and from these data, we could extract several parameters related to cellular bionenergetics. We initially assessed the change in the mean of OCR of CCs at P30 (Figure 7A) and P120 (Figure 7B) when exposed to oligomycin (0.5 µM), which blocks the ATP synthase; FCCP (2 µM), which collapses the proton gradient, disrupting the _m_ψ; and a mixture of rotenone (1 µM) and antimycin A (0.5 µM), which blocks complexes I and III, respectively. When the different parameters calculated were examined, the main differences between the WT and mutant cells were found in the following parameters: maximal respiration was 41.2% lower at P30 (* *p* < 0.05) and 24.7% lower at P120 (** *p* < 0.01); the spare respiratory capacity (SRC) was 148.6% lower at P30 and 37.4% lower at P120 (* *p* < 0.05); and the non-mitochondrial oxygen consumption was 44% lower at P30 (*** *p* < 0.001) and 59.7% lower at P120 (*** *p* < 0.001: Figure 7C). There were no significant differences in other parameters, such as basal respiration (Figure 7C(a)), proton leak (Figure 7C(b)), ATP production (Figure 7C(f)) and percentage of coupling efficiency (Figure 7C(g), see Table 1 for all the mean values and their standard errors).

#### 2.3.4. The SOD1^G93A^ Enzyme Is Expressed in the Chromaffin Cells of Transgenic Mice and Accumulates during Pathology Progression

An important aspect of our study was to demonstrate the expression of the mutated SOD1^G93A^ enzyme in CCs and to determine where it is located intracellularly, as well as any possible changes in this localization as the disease progresses. As such, immunofluorescence analysis was performed on CCs and Western blotting was carried out using samples from adrenal glands against the recombinant G93A human SOD1 mutant protein (hSOD1^G93A^). In confocal images of cultured CCs from WT mice and both P30 and P120 SOD1^G93A^ mice, punctate staining was observed in the SOD1^G93A^ CCs that was stronger at P120 (Figure 8A,B). Quantification of SOD1^G93A^ epifluorescence images showed a 3.1-fold increase in fluorescence at P30 relative to the WTs (*** *p* < 0.001), that augmented to 4.8-fold at P120 (*** *p* < 0.001: Figure 8C). Thus, there was a significant increase in SOD1^G93A^ immunoreactivity as the disease progressed and the motor symptoms were consolidated (^###^
*p* < 0.001). In Western blots using samples from the frozen adrenal medulla from five different mice, a protein band was only detected in the mutants, which was more intense in the P120 symptomatic mice (Figure 8D), with a 49% increase in the SOD1^G93A^ protein expression when the disease was established (P120) relative to the presymptomatic stages (Figure 8E).

#### 2.3.5. Mutated Human SOD1^G93A^ Is Located in Mitochondria, Accumulating More as the Disease Progresses

Having described both structural and functional alterations to the mitochondria from CCs of the SOD1^G93A^ mice, we performed experiments to explore the relationship between the mutated protein and the mitochondria. Since this type of association might be important in helping to demonstrate that the accumulation of SOD1^G93A^ is a possible cause of mitochondrial damage, we performed immunofluorescence on CCs from P30 and P120 SOD1^G93A^ mice (Figure 9), labeling mitochondria with MitoTracker Red, the mutated protein with the anti-human SOD1^G93A^ antibody (green) and the nucleus with DAPI (blue). In representative images, the mutated protein displayed a punctate cytosolic distribution similar to that described above (Figure 8), and we analyzed the colocalization of this signal with the mitochondrial staining, normalized to the total mitochondrial signal in a middle cross-section of the CC captured by confocal microscopy. At presymptomatic stages, the mean colocalization was 4.5 ± 0.8%, which increased to 10.9 ± 1.1% at symptomatic stages, almost tripling as the disease progressed.

To confirm the mitochondrial subcellular localization of the mutated enzyme, immunogold images of the same samples were used to characterize mitochondrial ultrastructure in the CCs, detecting the anti-human SOD1^G93A^ antibody with a 10 nm gold particle conjugated secondary antibody. The immunogold particles were localized to the intermembrane space (110.9 ± 37.0 particles/µm^2^), cristae (51.8 ± 32.6 particles/µm^2^) and mitochondrial matrix (24.9 ± 5.5 particles/µm^2^) at P120 (see Figure 10A for representative images from at P30 and P120 mice). Hence, hSOD1^G93A^ is located in mitochondria despite its cytosolic nature, and at the symptomatic stage, an average of 35.0 ± 21.8 immunogold particles/µm^2^ were detected in the mitochondria as opposed to 412.7 ± 76.2 particles/µm^2^ in the cytosol, consistent with the immunofluorescent colocalization indicated above. Furthermore, immunogold particles were also detected in different cellular compartments, such as the ER (where there was intense immunoreactivity), lysosomes, peroxisomes, different types of granules, the cytosol and within the nucleus (Figure 10B).

#### 2.3.6. The OPA1 Mitochondrial Fusion Protein Is Down-Regulated in Chromaffin Cells from SOD1^G93A^ Mice

The dynamic imbalance in mitochondria was studied through the key fusion protein OPA1. This protein is located at the inner mitochondrial membrane, and it is responsible for the final fusion of two independent mitochondria and for maintaining the structure of mitochondria and their cristae. The activity of this protein is controlled through protein cleavage, establishing the balance between the long (L-OPA1) and short (S-OPA1) isoforms. When the total OPA1 was analyzed in Eastern blots of the homogenized adrenal medulla from WT and from P30 and P120 SOD1^G93A^ mice (Figure 11), there was less OPA1 in the CCs of presymptomatic (25% lower; * *p* < 0.05) and symptomatic (36%; * *p* < 0.05) SOD1^G93A^ mice than in those from WT mice. We then analyzed OPA1 cleavage by quantifying the S-OPA1/L-OPA1 ratio, and this ratio changed with age due to the increase in S-OPA1 and the loss of L-OPA1, shifting from a ratio of 1 at P30 to a ratio of 1:1.37 at P120 (Figure 11C). Nevertheless, this age-related relative shift in these isoforms was not significantly different between the WT and transgenic mice (0.97 ± 0.04 and 0.92 ± 0.06 at P30, 1.37 ± 0.08 and 1.3 ± 0.07 at P120).

#### 2.3.7. Fewer Ubiquitinylated Proteins in Chromaffin Cells from the Adrenal Medulla of SOD1^G93A^ Mice, in the Absence of Changes in the Proapoptotic Bax Protein

The ubiquitin proteasome system (UPS) is the main resource in the cell used to eliminate proteins. To understand what occurs with this fundamental element of proteostasis in our ALS model during disease implantation, we analyzed the mono- and poly-ubiquitinylated proteins in the adrenal gland. Total protein from the adrenal medulla was probed via Western blots using the FK2 antibody, which recognizes ubiquitinylated proteins (see Appendix A). A smear of ubiquitinylated proteins was labelled in the adrenal medulla, which was weaker in the SOD1^G93A^ than in the WT medulla (24% at P30 and 22% at P120), although the difference was only significant in symptomatic P120 mice (* *p* < 0.05).

Apoptotic cell death is linked to ALS, and the interaction of the mutated human SOD1^G93A^ protein with the Bax protein is a well-documented interaction [33]. If this happens in CCs, and without the intention of doing an extensive study on apoptotic proteins, it could be a starting point for being able to explain our TEM images, in which we observed mitophagy processes (Figure 1B(h)). Thus, when Bax was assessed in Western blots of total protein from the adrenal medulla, there were no significant differences between the WT and the mutant, even in the symptomatic P120 mice, even though its expression diminished with age and disease progression (Appendix A).

## 3. Discussion

In this study, we used the SOD1^G93A^ mutant mouse model of ALS to assess the effects of this mutation on the mitochondria of CCs and how these might be related to the development of the disease.

### 3.1. Ultrastructural Damage to Mitochondria, as the First Event Described in ALS, also Occurs in Chromaffin Cells of SOD1^G93A^ Mice

One of the first events thought to affect motor neurons from mice carrying the G93A and G37R mutations in the SOD1 enzyme that develop ALS is the accumulation of a large number of small vacuoles around the membrane [5,6,7]. These small vacuoles (70–150 nm) derive from dilated mitochondria and ER, and they contain damaged cell components. This phenomenon is attributed to the acquired “toxicity” produced by the mutated enzyme. Specifically, the enzyme can exist in non-native conformations as a result of inappropriate folding (due to the mutation) or anomalous post-translational modification (e.g., oxidation: [34]). The protein has intrinsic toxic potential depending on the conformation [5,34] and on the specific mutation acquired, and its toxicity may be concentration-dependent [35]. Moreover, the subcellular localization of the mutated protein could influence the severity of the disease [36]. Interestingly, the misfolding of the quaternary structure appears to not only affect the SOD1 protein but it could also affect other proteins such as FUS, TDP43 or C9orf72, consistent with the premise that the development of the disease involves ultrastructural and functional alterations to mitochondria (reviewed in [37]).

We set out to determine whether CCs from the SOD1^G93A^ mutant mouse also experience the ultrastructure alterations to mitochondria previously described in motor neurons (e.g., [7,8,38]). We did detect the formation of small vesicles derived from mitochondria in CCs, as well as other notable changes, including the presence of dilated and disorganized cristae, evagination and fragmentation of the external mitochondrial membrane and the formation of mitophagosomes that contain mitochondrial material (Figure 1(h)). As in the motor neurons, vacuoles form through the progressive bloating of the intermembrane space. The expansion of the external mitochondrial membrane ultimately leads to the formation of large vacuoles and induces the collapse of the internal membrane [39,40]. Furthermore, it is notable that there is similar bloating of the internal mitochondrial membrane and expansion and vacuolization of the mitochondrial matrix.

### 3.2. Alterations to the MICOS System Affects the Ultrastructure of the Cristae Junctions

The internal mitochondrial membrane is divided into two membrane domains with distinct topologies and protein compositions. There is the inner boundary membrane, where the majority of the transporter proteins that access the internal mitochondrial compartments are located (e.g., translocase of the inner membrane: TIM), and the laminar membranes (also called cristae) that mainly contain the mitochondrial oxidative phosphorylation system (OXPHOS) protein complex and F_1_F_o_-ATP synthase. Both of these membrane domains are connected through the cristae junctions, which maintain the asymmetric composition of the mitochondria and restrict diffusion between these compartments. The nature of the cristae junctions is regulated by a multi-oligomeric protein complex known as a mitochondrial contact site and cristae organizing system (MICOS: [41]).

We find that while the opening of the cristae junction was less than 13 nm in the WT mice, it is double that size in SOD1^G93A^ mice at pre-symptomatic ages, and it increases 4-fold at symptomatic ages (Figure 4). The increase in the size of the cristae junctions augments as the disease progresses, reflecting an alteration to the MICOS system. Among the factors known to destabilize MICOS are alterations to the proteins that regulate this system, such as the Hsp70 chaperone [42] or mitochondrial dynamin-like GTPase OPA1 [43]. Another important factor is the peroxidation of cardiolipin by ROS, the principal phospholipid in the cristae junction membrane [44], the oxidation of which disrupts MICOS anchoring [44,45], an event known to occur in the SOD1^G93A^ mutant mouse [46]. Such alterations to MICOS affect mitochondrial fusion and the fragmentation of the cristae [43,47], which in turn lead to deficient mitochondrial respiration. These findings were also seen here, in the CCs of SOD1^G93A^ mice, as we explain next.

Finally, the opening of the cristae junction to the extent observed here, accompanied by the peroxidation of cardiolipin, may liberate large proteins into the intermembrane space (e.g., cytochrome c), where they may generate ROS and initiate apoptosis [46]. We studied the expression of Bax, a protein that regulates apoptosis through the liberation of cytochrome c and the activation of caspases, but failed to observe changes in its expression (see Appendix A). Although we cannot rule out a liberation of cytochrome c, TEM studies identified alterations to the mitochondrial ultrastructure, such as the loss of some content, the formation of vacuoles, or the liberation of ribosomes to the cytoplasm from the RER. These phenomena suggest that cell death may be produced by necrosis [48,49], a type of cell death previously associated with ALS [50].

### 3.3. Alterations to the Cristae Junctions and the MICOS System Affect the Mitochondrial Membrane Potential

We found that the SOD1^G93A^ mutant mice have a weaker electrochemical mitochondrial gradient (Figure 5). This alteration in the _m_ψ may reflect the tight relationship between the ultrastructural integrity of the mitochondrial cristae and the activity of the internal mitochondrial membrane via MICOS [41,51,52]. This may be due to some of the aforementioned factors, such as alterations to the proteins that regulate MICOS, such as OPA-1 [43], or alterations to cardiolipin [44,45,46]. These changes produce an increase in the activity of ion transporters and the internal mitochondrial membrane channels that provoke mitochondrial swelling, consequently affecting the _m_ψ [53]. Furthermore, the opening of the cristae junctions produces a diffusion of proteins, metabolites and ions from the mitochondrial cristae towards the intermembrane space, thereby increasing osmolarity and swelling [54]. As such, the morphology of the mitochondrial cristae determines the efficiency of respiration, affecting the _m_ψ [55].

### 3.4. Alterations to the _m_ψ, the Maximum Respiration and the Spare Respiratory Capacity Reflect the Inefficiency of the OXPHOS

We also observed a loss in basal _m_ψ in CCs from SOD1^G93A^ mice. It is logical to think that if the cell has a low basal _m_ψ, it will attempt to (i) avoid “*proton leak*” to try to maintain the proton-motive force (Figure 7C(b)), (ii) maintain its efficient coupling (Figure 7C(g)) and (iii) obtain reductive potential in the mitochondrial complexes in the form of NADH and FADH in order to maintain the proton gradient. Reductive potential can be provided by the TCA cycle and through the malate/aspartate or succinate/fumarate shuttle, given that these substrates do not depend so much on the _m_ψ for their entry into mitochondria. As such, the cell will maintain similar values of basal respiration (Figure 7C(a)) and ATP production (Figure 7C(f)), as in the WT mice, although this is at the price of excess expenditure in terms of metabolic substrates, which means an inefficient mitochondrial respiration. This is particularly evident when the cells of the mutant SOD1^G93A^ mice are subjected to an even greater collapse of the _m_ψ in the presence of FCCP. This forces cells to maximize oxygen consumption to compensate for the loss of the proton-motive gradient through an increase in electron transport and ATP production. In this situation, substrates such as ADP that were already compromised and that are transported by the adenine nucleotide transporters (ANTs) cannot be transported as they are dependent on the _m_ψ. Hence, the mutant SOD1^G93A^ mouse would be more dependent on this type of substrate due to its deficiencies in the basal _m_ψ. Indeed, we find alterations in maximal respiration (Figure 7C(c)) and in the spare respiratory capacity (Figure 7Cd) that are dependent on the _m_ψ, and consequently, a decrease in the ATP generation through OXPHOS is evident. It is important to note that the need to obtain reductive capacity means that, as metabolism is inefficient in the mutant mouse, a high glucose, protein and fat consumption is required to maintain the necessary NADH and FADH supply. This might explain why the SOD1^G93A^ mutant mice lose weight as the disease progresses (Appendix A). Of note, the weight loss in ALS has mainly been associated with muscle denervation and atrophy [56], and this has been also found in SOD1^G93A^ mice [57]. However, a situation of “mitochondrial hypermetabolism” should not be ignored to understand weight loss in ALS [58]. Additionally, this ADP availability impairment could explain our results of a greater adoption of the orthodox mitochondrial state in relation to the WT mice. In addition, it should also be noted that CCs have a more glycolytic than oxidative nature, whereby 20% of the glucose consumption is through glycolysis and 30% is required for lactate formation, more similar to the metabolism of a glial cell than a neuron [59]. Furthermore, if we consider the physiological changes in mitochondrial respiration produced with age, there is an increase in the maximal respiratory capacity of the WT mice and of their spare respiratory capacity (Figure 7C). These changes may be related to mitochondrial and cellular maturity, as reflected in the larger mitochondrial size and fusion (Figure 2). By contrast, in the SOD1^G93A^ mutant mouse, the maximal and spare respiratory capacity increases with age, but due to an increase in the number of small mitochondria, this respiratory capacity always being weaker than in the WT mice.

In summary, while alterations to the mitochondrial respiratory complexes have often been described in ALS patients [60,61] and in mouse models [62], the results presented here suggest that the alterations produced in OXPHOS in CCs are mainly the consequences of the reduced _m_ψ due to defects in MICOS.

### 3.5. The Disequilibrium in the OXPHOS System, the SOD1^G93A^ Toxic Gain-of-Function and the Alterations to Redox Homeostasis Are the Main Mechanisms That Generate Free Radicals

Although there are many potential sources of ROS, in CCs of the SOD1^G93A^ mice, their generation can be explained by four main mechanisms: (i) an imbalance in the OXPHOS system; (ii) toxicity of a misfolded SOD1^G93A^ protein; (iii) an effect on the enzymes that regulate redox homeostasis and (iv) the activity of other enzymes that generate ROS. The importance of each of these mechanisms are addressed below in the light of our data.

(i).Imbalance in the OXPHOS system

Under normal circumstances, between 2–6% of the oxygen consumed generates superoxide anions (O_2_^*−^ ). These superoxide anions are produced by a leakage of electrons to the mitochondrial matrix at the complexes I and III [63,64], which could produce electron transfer to oxygen through ubiquinone producing O_2_^*−^ [65]. Due to its origin in the electron transport chain, the production of mitochondrial ROS is tightly correlated with the _m_ψ. When the _m_ψ is high, the flux of electrons is slower, favoring the production of ROS. Likewise, the increase of ROS could be regulated by uncoupling the electron transport chain through various mechanisms: (a) dissipation of the proton gradient through uncoupling proteins (UCP_1–5_), known as “*proton leak*”; (b) through “*electron slip*” or the H^+^—electron stoichiometric variation that produces enhanced H^+^ transport for each electron. This loss of _m_ψ means that the final reaction of electron transfer to form ATP, CO_2_ and H_2_O is likely to be thermodynamically more spontaneous, minimizing the loss associated with the production of ROS [66]. As such, physiological factors such as age favor OXPHOS inefficiency, generating more ROS, as seen by the ROS production at both mouse ages (Figure 6). However, in the pathological conditions observed in the SOD1^G93A^ CCs, the decrease in the _m_ψ (Figure 5) arises without any change in “*proton leak*” or coupling efficiency, possibly due to a cell’s need to maintain the proton-motive gradient (Figure 7C). As indicated above, the cell will require extra reductive power through NADH and FADH to maintain the proton-motive gradient, and these sources will be incorporated into complex I and II respectively, increasing ROS production as a consequence of maintaining OXPHOS in conditions that drive a loss in _m_ψ [67].

(ii).Toxicity of the misfolded SOD1^G93A^ protein

It has been widely described that metal cations in the SOD1 enzyme complex can undergo oxidation-reduction reactions that cause O_2_^*−^ generation [68]. It is also known that the enzyme can release these transition metals, which can themselves catalyze the formation of ROS (Fenton reaction: [69]). Finally, misfolding and aggregation of SOD1 make it acquire new toxic properties [70].

(iii).Implication of the enzymes that regulate redox homeostasis

Cells have three main systems of antioxidant defense: the glutathione system, catalase enzymes and SOD1 [71]. Here, we find that the SOD1^G93A^ mutation produces incorrect folding and aggregation (Figure 9, Figure 10 and Figure 11), compromising the cells’ capacity to combat O_2_^*−^. It is important to note that the chemical reduction of all the aforementioned enzymes of the antioxidant defense system depends on NAD(P)H, which is mainly produced in the pentose phosphate pathway. Indeed, SOD1 needs the action of the NAD(P)H-dependent catalase to recover its activity. This metabolic pathway is of great relevance in CCs, with up to 5-fold greater activity in normal conditions relative to the nervous system [59,72] due to their need to synthesize proteins that are contained in their secretory granules [73]. However, the overexpression of SOD1^G93A^ can reduce the production of NAP(P)H by 20% [74], provoking a loss of these enzymes activity due to the lack of metabolic substrate.

In addition, gene expression of some transcription factors that participate in the regulation of oxidative stress, such as Nrf2, may be repressed, as may be that of the α, μ and π of glutathione transferase isoforms [74,75,76]. Taking this information into account, the increase in oxidative stress might be associated not only with an increase in ROS production but also with an impaired antioxidant cell defense. This would be aggravated in those cells that have not developed a capacity for antioxidant defense due to their oxidative metabolism, as is the case of human CCs [59]. In these cells, for example, the π isoform of glutathione transferase is expressed moderately, while other isoforms are lacking, such as the α isoform [77,78].

(iv).Other enzymes that generate ROS

There are other enzymes that are very active in terms of generating ROS, the expression of which is generally inducible, such as NAD(P)H oxidase (NOX), xanthine oxidase, cytochrome P450 or inducible nitric oxide synthase (iNOS). An increase in their expression is often triggered by damage, and it leads to high oxygen consumption at the expense of its activity. Our results show that CCs have low non-mitochondrial oxygen consumption (Figure 7C), and thus the activity of these enzymes cannot explain the increase in ROS observed (Figure 6), suggesting that they are not the main cause of the high oxidative stress that occurs in the SOD1^G93A^ mouse. In fact, some studies have suggested that the relationship between ROS generation and NOX expression is not proportional (an increase over 20-fold in a subunit of the NOX protein complex producing only 1.5-fold increase in ROS: [79]).

Finally, it is important to note that the pathological levels of ROS observed here could damage nucleic acids, proteins (carbonylation or nitration) and lipids (peroxidation), as described widely in the SOD1^G93A^ mouse [80]. Certain by-products of such reactions, such as those derived from protein glycation, can evolve into advanced glycation end products and exaggerate the effects of ROS [81,82], which should also be taken into consideration.

### 3.6. The Mitochondrial Alterations Observed Are Caused by Protein Modifications of the MICOS Complex That Participate in Mitochondrial Homeostasis, Such as OPA1

Mitochondria can undergo fusion and fission events, processes that are highly regulated to control mitochondrial damage. These processes permit degradation of damaged parts of the mitochondria by mitophagy, or the dilution of the damage by mitochondrial fusion [83]. Mitochondrial fusion and regulation of the morphology of the mitochondrial cristae is controlled though specific proteins, including OPA1 localized to the mitochondrial inner membrane [52]. The human mitochondria contain at least eight isoforms of OPA1 generated by alternative splicing. Its proteolytic cleavage by the YME1L and OMA1 proteases will also produce the “long forms” (L-OPA1) and “short forms” (S-OPA1) of this protein. The correct balance between these two forms exerts a reciprocal control over mitochondrial morphology [84].

Alterations to mitochondrial ultrastructure were evident in the present study, with a swelling of the crista, the intermembrane space and the matrix in the CCs of the SOD1^G93A^ mutant mice (Figure 1). In addition, the cells of the mutant SOD1^G93A^ mice had increasingly more but smaller mitochondria (Figure 2). Their cristae were larger due to swelling and their number gradually fell (Figure 3), with the cristae junctions maintaining a larger opening as the disease progressed (Figure 4). By contrast, the control mice had increasingly fewer but larger mitochondria with aging, maintaining a proportional relationship between the increase in mitochondrial size and their cristae. These changes suggest that mitochondrial fusion may be deficient in the SOD1^G93A^ mutant mice, and because the mitochondrial morphology is dependent on OPA1, this protein was studied in more detail.

One of the most relevant findings of this study is that there is a greater clearance of total OPA1 in the SOD1^G93A^ mouse (Figure 11), even at presymptomatic ages, consistent with the changes in mitochondrial ultrastructure. On the other hand, the S-OPA increases as L-OPA1 decreases in both mice, which can be explained physiologically by the reciprocal regulation by the YME1L and OMA1 proteases as a consequence of the variation of oxidative stress and cellular ATP concentrations with age [85]. In addition, the more intense mitochondrial depolarization observed in the SOD1^G93A^ mutant mice (Figure 5), in conjunction with deficiencies in OXPHOS (Figure 7), should stabilize OMA1 and prevent mitochondrial fusion through OPA1 clearance [86]. However, the fact that there is a decrease in the expression of total OPA1, but with no change in the S-OPA1/L-OPA1 ratio between the control and the SOD1^G93A^ mice, suggests that changes in total OPA1 occur through regulating gene expression and not only through post-transcriptional regulation. In this sense, it is notable that the decrease in OPA1 expression is conditioned by the increase in ROS production in several animal models [87,88]. This link is due to ROS repressing the expression of OPA1 through the action of certain transcription factors, such as the kappa-light-chain-enhancer of activated B cells factor (NF-κB: [89,90,91]). This relationship could explain our observations, although further studies will be necessary to define the mechanisms regulating OPA1 gene expression.

Finally, in addition to regulating OPA1 and inducing mitochondrial fragmentation, NF-κB can activate the PINK1/ubiquitination-parkin-dependent pathway, which results in the elimination of damaged mitochondria by mitophagy [90,91]. We observed mitophagosomes in the CC of SOD1^G93A^ mice (Figure 1B(h)), and thus, NF-κB may participate in this process. Other aforementioned factors may also contribute to this process, and disruption of the MICOS machinery can influence the expression of OPA1 [43]. Therefore, multiple mechanisms may be at play that regulate OPA1.

### 3.7. The Accumulation and Aggregation of SOD1^G93A^ in the Mitochondria Alters the Proteins Involved in Mitochondrial Homeostasis

The accumulation of the misfolded SOD1^G93A^ protein in the mitochondria has been proposed as a possible explanation for the degeneration of motor neurons in ALS [92]. Spinal cord and motor neurons accumulate the mutant SOD1, and they are particularly sensitive to the resulting mitochondrial dysfunction [8,62,93,94,95]. We verified that the expression of the mutant protein increases in parallel with disease progression in the CCs of the SOD1^G93A^ mouse (Figure 8). More specifically, using confocal microscopy techniques to assess colocalization (Figure 9) and TEM immunogold labelling of the SOD1^G93A^ protein (Figure 10), we verified the parallelism between the accumulation of aggregates at the cellular and mitochondrial level with age. We also observed that SOD1^G93A^ protein accumulates in the intermembrane space, as well as in the mitochondrial cristae and the matrix. These results have not previously been described in such depth, and the mutant enzyme was mainly circumscribed to the mitochondrial intermembrane space, attached to the outer mitochondrial membrane and interacting with voltage-dependent anion channels (VDACs: [96]) or Bcl-2 [97]). Indeed, SOD1^G93A^ is found in all mitochondrial structures, and its accumulation is closely related to the alterations described in the mitochondrial function. Moreover, the mutant protein could also accumulate in non-mitochondrial sites, such as the ER, nucleus, cytoplasm (Figure 10E,F), lysosomes, peroxisomes and small granules or cisterns (data not shown). Although its presence in some of these locations has been described previously [98]; interestingly, its specific localization in the ER is associated with pathological interaction with chaperones. Activation of them induces reticular stress and causes neuronal death mediated by caspases in the transgenic SOD1^G93A^ mice [99,100].

In eukaryotic cells, the mitochondrial localization of SOD1 is regulated by its folding. Once synthesized in the cytoplasm by ribosomes, the unfolded form of the SOD1 enzyme enters the mitochondria through the transporter of outer membrane (TOM: [101]) or the mitochondrial intermembrane space through the MICOS [102]. The mitochondrial distribution and protein maturation of SOD1 is mediated by a copper chaperone that is responsible for the formation of the SOD1 complex with Zn^2+^ and for inducing conformational rearrangements through disulphide bonds (for review see: [103]). Once folded, the monomers of SOD1 dimerize to form the active enzyme. The presence of misfolded SOD1^G93A^ proteins may lead to oligomerization between native and aberrant SOD1 proteins, causing their aggregation and their ensuing trapping in the mitochondrial intermembrane space, as they are not transported in their native mature form. However, an important finding of our study is that SOD1^G93A^ can also be localized to the matrix and mitochondrial cristae, which implies that there has to be an immature conformation that allows it to be transported to these locations. As far as we know, the presence of immature aggregates in the mitochondrial matrix of motor neurons has only been described in SOD1^G93A^ and SOD1^G85R^ mice [104]. In addition, the fact that the accumulation of the SOD1^G93A^ protein increases with time and correlates with the extent to which the cristae junctions open suggests that this mechanism allows migration of the protein to the mitochondrial cristae, an event not previously described.

It is important to highlight that other mechanism such as NF-κB may also regulate OPA1. In the context of the SOD1^G93A^ accumulation described here, it is important to consider the heat shock proteins (HSPs) that act as chaperones that recognize misfolded proteins, helping to keep them in their correct conformation or directing them to the ubiquitin–proteasome pathway for degradation [105]. In fact, the 40 and 70 kDa HSP chaperones (Hsp40 and Hsp70) recognize the SOD1^G93A^ mutant protein [106], and their overexpression prevents motor neuron death [107,108]. The excessive SOD1^G93A^ accumulated may sequester Hsp70, preventing its degradation in the ubiquitin–proteasome pathway [109,110]. Interestingly, Hsp40 and Hsp70 also fulfil important roles in modulating mitochondrial morphology through the clearance of the OPA1 protein [111]. Although we did not measure the expression of Hsp40 and Hsp70, the decrease in the total OPA1 (Figure 11) and the loss of ubiquitin–proteasome pathway activity evident with the FK2 antibody that recognizes mono- and poly-ubiquitinylated proteins (Figure 12), leads us to speculate that SOD1^G93A^ may interact at least with the mitochondrial chaperone Hsp70, potentially contributing to the alterations in mitochondrial morphology. Therefore, multiple mechanisms could affect OPA1 or alter proteins that participate in mitochondrial homeostasis. It is necessary to better understand the involvement of these proteins in the mitochondrial alterations in order to decipher the underlying causes of ALS. Studies of osteocytes cell lines and those derived from the SOD1^G93A^ mouse recently suggested that the mutant enzyme could alter mitochondrial morphology through OPA1 [112]. The importance of our study is that it is the first that proposes this mechanism in a non-motor nerve cell, and we discuss the relevance of the changes in the context of morphology, metabolism and mitochondrial function. Studying the effects of the SOD1^G93A^ mutant enzyme in different cell types (such as glia, osteocytes, hepatocytes or CCs) sheds light on the importance of the metabolic phenotype in cell survival and on the damage occurring in ALS.

## 4. Materials and Methods

### 4.1. Animals

In this study, ALS mice (B6.Cg-Tg(SOD1*G93A)1Gur/J) were used that overexpressed a point mutated form of the human SOD1 gene (chromosome 12, substitution of glycine to alanine at codon 93) driven by the endogenous human promoter. The colony was maintained by strict crossing of the hemizygous male SOD1^G93A^ transgenic mice (kindly provided by Professor Josep E. Esquerda, Universitat de Lleida) with wild-type (WT) C57BL/6J inbred females in order to maintain the mutant transgene on a C57BL/6 congenic background. The offspring were genotyped to select the SOD1^G93A^ positive transgenic mice (SOD1^G93A^) and the SOD1^G93A^ negative mice as WT. Mice were housed under controlled conditions: 22 ± 2 °C, 60 ± 20% relative humidity, 12 h dark/light cycle, and food and water ad libitum. Experiments were performed on presymptomatic mice euthanized on P30–45 (called P30 throughout post-natal day 30) or symptomatic mice euthanized on P120–140 (called P120 throughout). Mice were sacrificed by cervical dislocation after deep anesthesia by CO_2_ inhalation before removing the tissues required.

### 4.2. Body Weight and Rotarod Test

The body weight of the SOD1^G93A^ and WT mice in each group was assessed after euthanized. The rotarod test was used to assess motor performance at P30 and P120 on a Roto-Rod series 8 apparatus (Stoelting, Wood Dale, IL, USA). Mice were placed on the rolling tube at an initial speed of 8 rpm, increasing by 1 rpm every 8 s until they fell off (120 s cut-off). The latency to fall was measured in seconds, averaging 10 repetitions for each animal tested (See Appendix A). Mice were trained for 1 day before the test was performed to habituate them to the Rota-Rod apparatus.

### 4.3. Cell Culture

Adrenal glands were isolated from the mice under sterile conditions, decapsulated under a dissecting microscope, and the medulla was carefully dissected out with a scalpel. The adrenal medulla was dissociated enzymatically with 25 U/mL papain (20 min) in a tube containing 200 µL Locke´s solution adjusted to pH 7.4 with NaOH (in mM): 154 NaCl, 5.5 KCl, 3.6 NaHCO_3_, 10 HEPES and 5.5 D-glucose. Enzymatic digestion was stopped by washing the cells three times in 1 mL Dulbecco’s Modified Eagle’s Medium (DMEM) with 8% fetal bovine serum (FBS), resuspending the tissue in a final volume of 100 µL. The medullae were dissociated mechanically with a 1 mL micropipette, using then a 200 µL micropipette and finally, with a 10 µL micropipette to disaggregate the CCs. Drops of the homogenate (10 µL) were plated on poly-D-lysine-coated glass coverslips in 9-, 12- or 48-well plates. After 30 min at 37 °C in a water saturated and 5% CO_2_ atmosphere to allow the cells to attach to the dish, 2 mL DMEM was added to each well (supplemented with 4% FBS, 50 IU/mL penicillin and 50 µg/mL streptomycin), and the cells were cultured for 24 h. The experiments were performed 1 day after plating.

### 4.4. Transmission Electron Microscopy (TEM) Images

Adrenal glands from three WT and three SOD1^G93A^ mice at P30 and P120 were dissected and fixed for 8 h at 4 °C by immersion in 4% paraformaldehyde, 1% glutaraldehyde, prepared in 0.1 M cacodylate buffer (pH 7.4). The adrenal glands were then cryoprotected, and coronal blocks (1.2–2 mm, 250 µm) from each adrenal gland were then fixed for 2 h at 4 °C in 2% osmium tetroxide buffered with 0.1 M Sorensen phosphate buffer (0.133 M Na_2_HPO_4_, 0.133 M KH_2_PO_4_, pH 7.2, 320 mOsm). The samples were then dehydrated by passing them through increasing alcohol concentrations (30, 50, 70 and 95% *v*/*v*, once for 10 min each), absolute ethanol (twice, 10 min each) and acetone (twice, 5 min). They were then embedded in Araldite resin (Durcupan ACM; Sigma, Madrid, Spain). Sections were obtained with diamond knives on a Leica Ultracut S^®^ ultramicrotome, mounted on a 200-mesh grid and stained with uranyl acetate and lead citrate. Micrographs were obtained on a JEOL transmission electron microscope JEM1010 (80 KV) using a digital camera (Gatan^®^, Orius 200 SC model) at an amplification of 5000× for whole cell images or 40,000× for detailed mitochondrial images.

### 4.5. Mitochondrial Ultrastructure and Morphometric Analysis

The mitochondrial area, matrix, cristae and intermembrane space were measured in 83 micrographs of mitochondria (17 WT and 28 SOD1^G93A^ at P30 and 19 WT and 19 SOD1^G93A^ at P120) acquired at 40,000× magnification using ImageJ software (v1.52b, National Institutes of Health, MD, USA). After image calibration on the scale (1 µm = 470 pixels), the different mitochondrial membranes were selected with the freehand selection tool and all parameters were measured with the region of interest (ROI) manager tool of the ImageJ software.

### 4.6. Immunoelectron Microscopy

Ultrathin sections of adrenal glands were etched in 0.1 N HCl for 5 min, rinsed three times for 5 min each in Tris buffered saline (TBS: 20 mM TRIS, 140 mM NaCl, and 2.7 mM KCl, pH 8.0), treated with blocking buffer (0.1% gelatin, 1% normal goat serum (NGS) and 0.3% Triton X-100 in TBS) for 30 min, incubated for 2 h at room temperature with the primary antibody (anti-hSOD1^G93A^, 1:500; Medimabs, C4F6, Montréal, QC, Canada). The sections were then rinsed three times for 5 min each in TBS, exposed for 1 h to a 10 nm gold-conjugated secondary antibody (anti-mouse IgG), rinsed three times in TBS, rinsed in water, stained with Reynold’s lead citrate followed by aqueous 2% uranyl acetate and then dried on filter paper.

### 4.7. Monitoring the Production of Intracellular Reactive Oxygen Species (ROS)

CCs were seeded in 6-well plates, as described previously, and cultured for 24 h. Cells were incubated for 30 min (37 °C, water saturated, 5% CO_2_ and dark conditions) in DMEM containing 26 μM of the ROS-sensitive probe 6-chloromethyl-2’,7’-dichlorodihydrofluorescein diacetate, acetyl ester (CM-H_2_DCFDA; LifeTechnologies^®^, MA, USA). Subsequently, the coverslips were mounted in a chamber where the cells were washed and covered with Tyrode’s solution (in mM): 1.37 NaCl, 5.3 KCl, 2 CaCl_2_, 1 MgCl_2_, 10 HEPES and 10 D-glucose, pH 7.4. The production of ROS by individual cells at rest was continuously monitored at 1 min intervals under a microscope over 30 min, selecting each cell with a different ROI. The set-up for fluorescence involved an inverted light microscope (DMI 4000 B; Leica Microsystems^®^, Barcelona, Spain) equipped with an oil immersion objective (Leica ×40 Plan Apo, numerical aperture 1.25). CM-H_2_DCFDA was excited at 472 ± 30 nm with a Leica EL6000 external light source and a Kübler CODIX Mercury lamp. The fluorescence emitted was collected through a 520 ± 35 nm emission filter and measured with an intensified charge-coupled device (CCD) camera (camera controller C10600 orca R2; Hamamatsu, Japan). Images were stored digitally, and the mean fluorescence intensity of each cell was analyzed using LAS AF software (Leica Microsystems^®^).

### 4.8. Mitochondrial Membrane Potential (_m_ψ)

Cultured CCs were incubated for 5 min with 25 nM tetramethylrhodamine ethyl ester perchlorate (TMRE; Molecular Probes^®^, MA, USA), and the _m_ψ was calculated by comparing the difference in fluorescence after depolarizing stimuli with the mitochondrial uncoupler carbonyl cyanide 4-(trifluoromethoxy) phenylhydrazone (FCCP, 3 µM). The TMRE probe was excited at 545 ± 30 nm and recorded through a 610 ± 75 nm emission filter. Images were stored digitally, and the ROIs of mitochondrial staining were analyzed using LAS AF software.

### 4.9. Bioenergetic Assay

A Seahorse XFp cell analyzer (Agilent Technologies^®^, Santa Clara, CA, USA) was used to study the bioenergetics of cultured CCs. This equipment allows measurement of the basal OCR in cells, also during pharmacological modulation of mitochondrial activity. We used the commercial Mito Stress Test Kit to study the CCs plated in an 8-well Seahorse plate and cultured as described previously. Each well corresponded to a culture from an individual mouse, and different plates were used for each age studied, P30 and P120. The following day, the medium was replaced with Agilent Seahorse XF Base Medium (Agilent Technologies^®^) supplemented with glutamine (2 mM), pyruvate (1 mM) and glucose (10 mM), and the cells were incubated at 37 °C in a non-CO_2_ incubator for 1 h. The sensor cartridge (previously hydrated overnight at 37 °C in Agilent Seahorse XF Calibrant in a non-CO_2_ incubator) was loaded with the drugs for the standard assay following the manufacturer’s instructions, obtaining final concentrations in the well of 1 µM oligomycin, 1 µM FCCP and 0.5 µM rotenone/antimycin A. After calibration of the sensors, the plates with the cells were placed in the Seahorse XFp analyzer and the Mito Stress Test was initiated. This standard test analyzed the OCR and extracellular acidification rate (ECAR) three times in a basal state and after each drug addition. Once the test had terminated, the cells were fixed with methanol and stained with 4′,6-diamidino-2-phenylindole (DAPI) to count the nuclei, and the experimental data was normalized to the cell number.

### 4.10. Immunofluorescence

Cells cultured for 24 h were treated for 45 min with 500 nM of the mitochondrial probe MitoTracker™ Red CM-H2Xros (Molecular Probes^®^, Life Technologies^®^, MA, USA) in serum-free DMEM medium in the incubator. After washing in PBS (phosphate buffered saline, in mM: 137 NaCl, 2.7 KCl, 10 Na_2_HPO_4_ and 1.8 KH_2_PO_4_, pH 7.4), the cells were fixed for 20 min in 4% paraformaldehyde. They were then washed twice with PBS before being permeabilized and incubated for 1 h at room temperature with blocking buffer (0.1% Triton X-100, 5% NGS in PBS). The cells were then incubated at 4 °C overnight with the primary antibody against misfolded human SOD1^G93A^ protein (1:500 dilution; Médimabs, C4F6, Montréal, QC, Canada) and then, after two washes in PBS, antibody binding was detected for 1 h at room temperature in the dark with the secondary Alexa Fluor 488 goat anti-mouse antibody (1:1000 dilution; Molecular Probes^®^). Both antibodies were diluted in PBS containing 0.1% Triton X-100 and 1% NGS. The cell nuclei were counterstained for 15 min at room temperature in the dark with DAPI (1 μg/mL, Molecular Probes^®^) and then mounted with liquid ProLong^®^ Gold Antifade medium (Molecular Probes^®^, LifeTechnologies^®^, MA, USA). Epifluorescence images were captured on an inverted light microscope (DMI 4000 B; Leica Microsystems, Barcelona, Spain) equipped with an oil immersion objective (Leica x40 Plan Apo, numerical aperture 1.25). The hSOD1^G93A^ signal (green) was collected with a 472 ± 30 nm excitation and 520 ± 35 nm emission filter (green fluorescent protein, GFP), and the DAPI nuclear blue signal was collected with a 360 ± 40 nm excitation and a 470 ± 40 nm emission filter. Confocal images were captured on a confocal ZEISS LSM 800 microscope equipped with an oil immersion Plan-Apochromat 63× magnification objective using the ZEN software (Carl Zeiss AG^®^, Germany). To stain the F-actin cytoskeleton, cells were incubated for 45 min with Alexa Fluor 546 Phalloidin (5:200) prior to DAPI staining.

### 4.11. Western Blots

Frozen adrenal medulla tissue was analyzed via Western blotting after homogenization in 100 µL M-PER^®^ Mammalian Protein Extraction Reagent (Thermo Scientific^®^, MA, USA) supplemented with protease inhibitor cocktail (Sigma-Aldrich^®^, MA, USA) using an ultrasonic processor UP50H (Hielscher Ultrasonics GmbH^®^, Teltow, Germany). The protein content was quantified with a Bicinchoninic Acid Protein Assay (BCA; G-Biosciences, MO, USA), and then 30 µg of total protein was resolved by electrophoresis on a 12% sodium dodecyl sulfate–polyacrylamide gel (SDS–PAGE) and transferred to a nitrocellulose membrane Immobilon-P^®^ Transfer Membranes (Millipore Corporation, MA, USA). Membranes were blocked for 1 h in TBS containing 0.1% Tween 20 and 5% milk, and they were then incubated overnight at 4 °C with primary antibodies against optic atrophy protein-1 (OPA1, 1:1000; Cell Signaling^®^, #80471, MA, USA), Bax (1:1000; Cell Signaling^®^, #2772), hSOD1^G93A^ (1:500; Médimabs^®^, C4F6) or ubiquitinlyated proteins (1:1000; FK2 Merck Millipore^®^, #04–263). Antibody binding was detected by incubating for 1 h with peroxidase conjugated secondary antibodies (1:5000; Santa Cruz Biotechnology^®^, TX, USA) and visualized with a Fujifilm LAS-4000 system (Fujifilm^®^, Tokyo, Japan) using ECL Select^®^ Western Blotting Detection Reagent (RNP2232, GE Healthcare^®^, IL, USA). A conjugated antibody against β-actin (ACTB, 1:50000; Sigma-Aldrich^®^) was used as a loading control, and the different band intensities for the proteins detected were quantified using ImageJ Software.

### 4.12. Statistical Analysis

The analyses were performed from at least three adrenal medulla from three different mice per group. The data are given as the mean ± SEM for the number (*n*) of mice/cells/mitochondria/cristae, analyzed using the GraphPad Prism^®^ version 9 for iOS (GraphPad Prism^®^ Software, San Diego, CA, USA). For all variables measured, a D’Agostino–Pearson omnibus normality test was applied. When it was indicated, the data fitting a normal distribution were compared using a parametric unpaired two-way Student’s *t* test, while data that did not follow a normal distribution were analyzed with a Mann–Whitney test. When more than two statistical variables were compared, a two-way ANOVA with Šídák’s multiple comparison post hoc test was performed. The limit of significance was set at * *p* ≤ 0.05, and ** *p* ≤ 0.01 and *** *p* ≤ 0.001 was taken as statistically significant. At least four individual experiments were performed for every assay, each with no fewer than three replicates.

## 5. Conclusions

The aim the study presented here was to examine the main mitochondrial alterations observed in CCs of SOD1^G93A^ transgenic mice at different stages of disease development: asymptomatic and symptomatic. The studies carried out focused on mitochondrial ultrastructure, metabolism and function, to better understand the mechanisms involved in ALS and the causes underlying the progression of ALS.

We show that the mutated SOD1^G93A^ protein accumulates inside the mitochondria in the intermembrane space, cristae and matrix. At these sites SOD1^G93A^ (i) impedes mitochondrial fusion and maturation with age; (ii) provokes ultrastructure changes such as swelling, vacuolization, loss of cristae and exaggerated opening of the cristae junctions; (iii) drives the loss of _m_ψ, the generation of ROS and metabolic alterations in OXPHOS; (iv) dampens the activity of the ubiquitin–proteasome pathway and (v) causes mitophagy. These events occur at an early age (disease stage) and are directly associated with the degree of intra-mitochondrial expression of the mutated SOD1^G93A^ enzyme, deteriorating as the disease progresses. Our main observation is that SOD1^G93A^ causes these mitochondrial alterations through the regulation of OPA1 gene expression. Moreover, while multiple pathways may be involved in this regulation, our results allow us to suggest that chaperones including Hsp70 or transcription factors such as NF-κB may also participate in this process. Further detailed studies should be carried out to better define these specific mechanisms.

The fact that the SOD1^G93A^ protein accumulates and induces mitochondrial damage in CCs implies some impairment of the sympathoadrenal axis. The clinical relevance of our study is that patients may be poorly adapted to physiological stress (for example, when practicing intense exercise) or pathological situations (for example, catecholamine secretion in response to ischemia). Indeed, ALS patients die as a result of sudden death syndrome because they are unable to compensate for cardiorespiratory arrest. Thus, the findings presented here may help to identify new therapeutic targets on which to act pharmacologically, such as OPA1 or Hsp70.

## Figures and Tables

**Figure 1 ijms-22-08194-f001:**
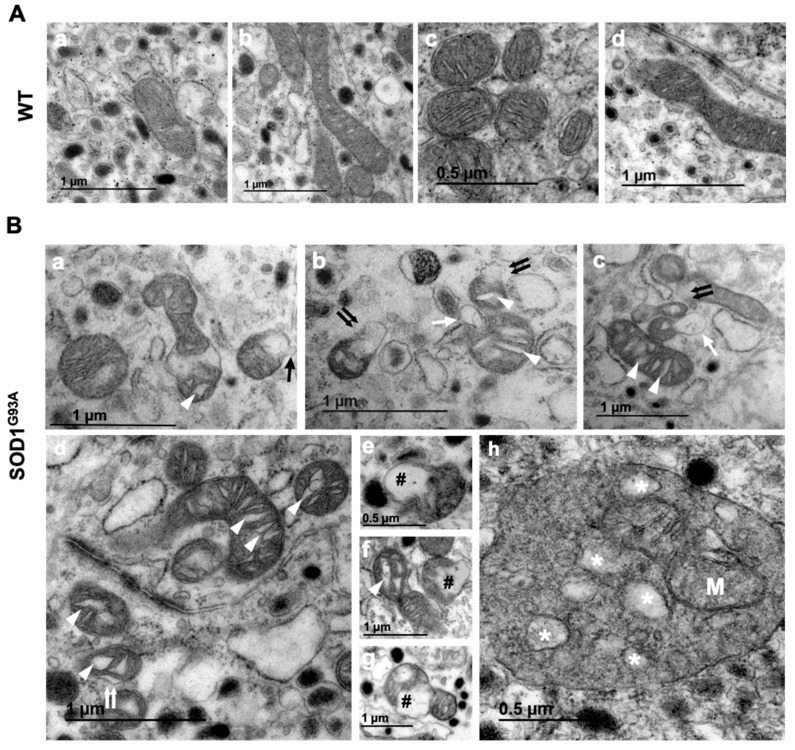
Mitochondrial damage in chromaffin cells of SOD1^G93A^ mice. (**A**) Transmission electron microscopy images showing a representative example of an orthodox mitochondria in CCs from control mice (WT) at both time points, with a circular, elliptical, tubular or branched shape: P30 (**Aa**,**Ab**) and P120 (**Ac**,**Ad**). Note the abundance of dense core granules containing catecholamines all around the cytosol, a hallmark of this cell type. WT mitochondria have fairly regular spaced lamellar cristae. (**B**) Microphotographs from SOD1^G93A^ mouse, showing the different mitochondrial alterations identified at both stages: mitochondrial sprouting of translucent vesicles ((**Ba**); black arrow), mitochondrial vacuoles derived from the matrix or cristae ((**Bb**,**Bc**); white arrow), mitochondrial outer membrane expansion ((**Bd**); double white arrow), cristae swelling ((**B**), white arrow head), disruption of the mitochondrial membrane ((**Bb,Bc**); double black arrow), mitochondrial matrix dilatation ((**Be**–**Bg**), black pad #) and mitochondrial degradation (M) by mitophagy ((**Bh**), degenerating organelles are marked with an white asterisk *). All images were obtained from adrenal gland tissue from three different mice per group.

**Figure 2 ijms-22-08194-f002:**
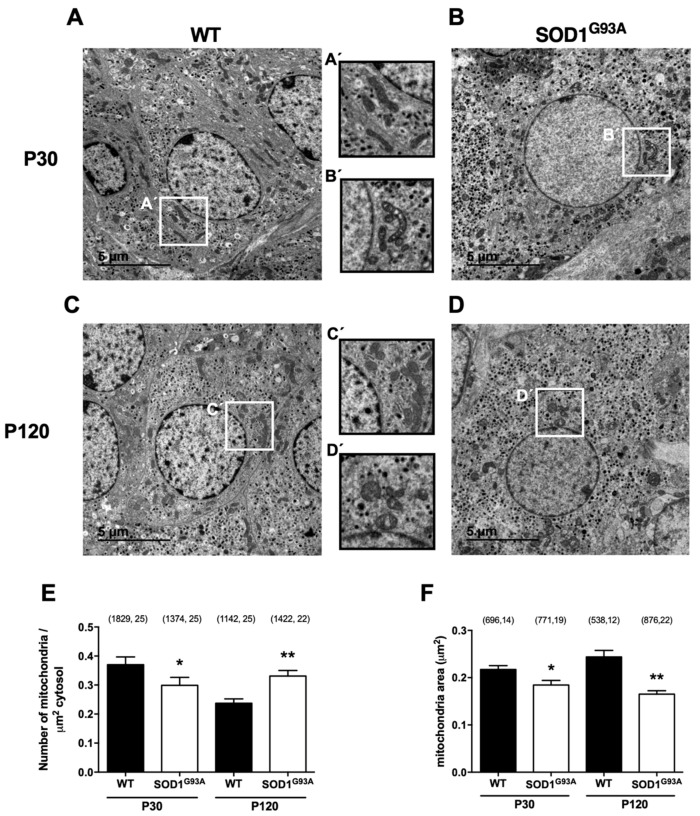
Changes in mitochondrial number and size serves as evidence for the dynamic impairment in chromaffin cells of SOD1^G93A^ mice. TEM images (5000×) obtained from the adrenal medulla of WT (left) and SOD1^G93A^ (right) mice. Each image is a cross section of a CC at presymptomatic P30 (**A**,**B**) or symptomatic P120 stages (**C**,**D**). (**A’**–**D’**) are magnifications of a cytoplasmic area with a high density of mitochondria (inset in each image). Note the internal alterations in the mitochondria from SOD1^G93A^ mice compared with the WT at both ages. (**E**) Average number of mitochondria in CCs normalized to the area (μm^2^) of cytosol (the number of mitochondria counted and pictures analyzed are shown in parentheses: * *p* < 0.05 and ** *p* < 0.01 comparing WT with SOD1^G93A^ at the same age). (**F**) bar graph summarizing the average data of the mitochondrial area (μm^2^): * *p* < 0.05; ** *p* < 0.01. Data are the mean ± SEM. The analyses were performed from at least three adrenal medulla from three different mice per group. ^#^
*p* < 0.05 comparing the effect of aging in the WT group and in the SOD1^G93A^ group. Statistical analyses were performed with two-way ANOVA with Šídák’s multiple comparison post hoc test.

**Figure 3 ijms-22-08194-f003:**
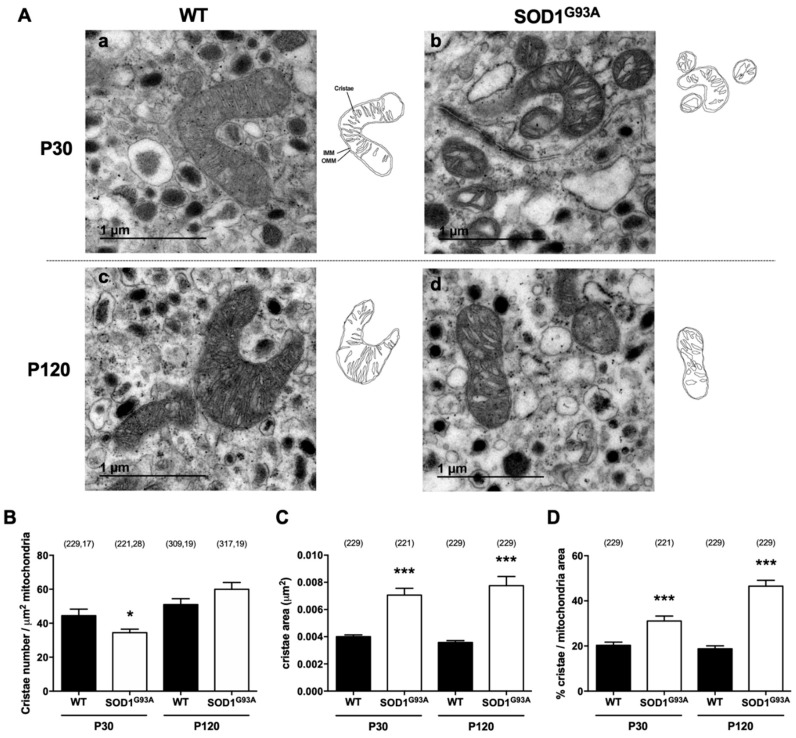
Mitochondrial cristae swelling in SOD1^G93A^ chromaffin cells. (**A**) TEM images (40,000×) acquired from adrenal medullae tissue from WT and SOD1^G93A^ mice showing independent mitochondria at presymptomatic (P30, (**a**,**b**)) or symptomatic stages (P120, (**c**,**d**)). Next to each picture there is a representative drawing of the mitochondria analyzed, achieved by manually selecting the inner and outer mitochondrial membranes and the cristae. (**B**) Bar graph depicting the average number of cristae per μm^2^ of mitochondria (between 200 and 300 cristae were measured per group). (**C**) Quantitative analysis of the mean cristae size (μm^2^). The mitochondrial area filled by cristae is represented as a percentage in (**D**). Data represent the mean ± SEM of the number of cristae and mitochondria (in parentheses) from at least three mice. The statistical analysis was performed using a two-way ANOVA with Šídák’s multiple comparison post hoc test: * *p* < 0.05; *** *p* < 0.001 vs. WT at the same stage. ^###^
*p* < 0.001 comparing the effect of aging in the WT and SOD1^G93A^ groups (only in panel B).

**Figure 4 ijms-22-08194-f004:**
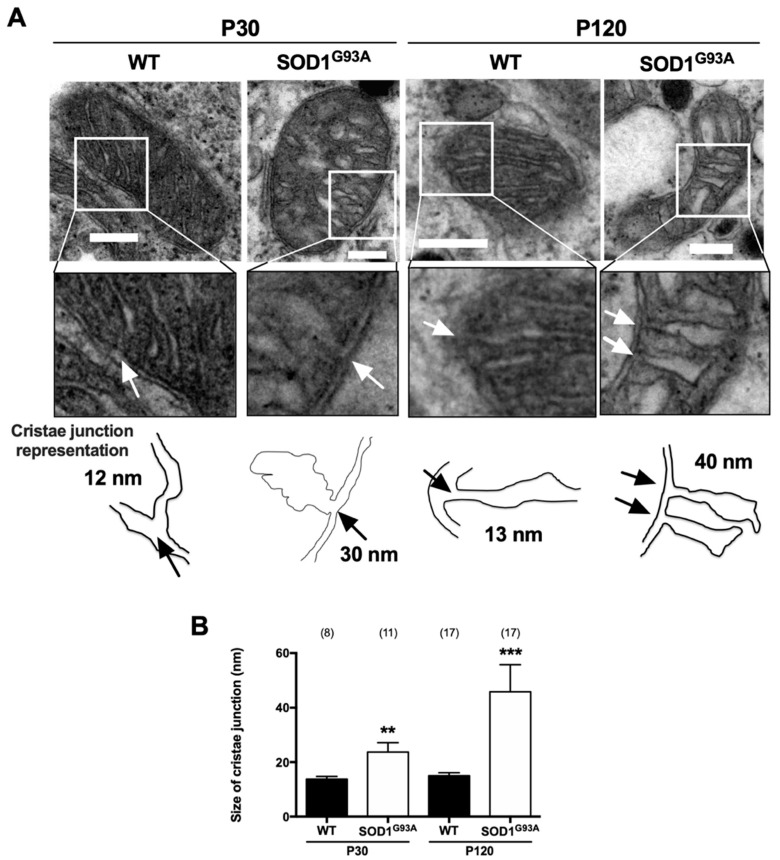
Widening of the cristae junctions in the mitochondria from SOD1^G93A^ mice chromaffin cells at both presymptomatic and symptomatic stages. (**A**) Cross section of mitochondria obtained by TEM from WT and SOD1^G93A^ mice at presymptomatic (P30) or symptomatic (P120) stages (40,000×). Magnifications of the cristae junctions observed are shown below each picture (see arrow) and at the bottom of the panel; a drawing of the mitochondrial external membrane and the cristae junction of each image can be found, with the cristae junction size in nm. (**B**) Bar plot of the average cristae junction size (in nm) in all groups. Data represent the mean ± SEM, and the statistical analysis was performed using the two-way ANOVA with Šídák’s multiple comparison post hoc test: ** *p* < 0.01 and *** *p* < 0.001 vs. WT at the same stage. Scale bar = 200 nm. There was no difference with respect to variable age.

**Figure 5 ijms-22-08194-f005:**
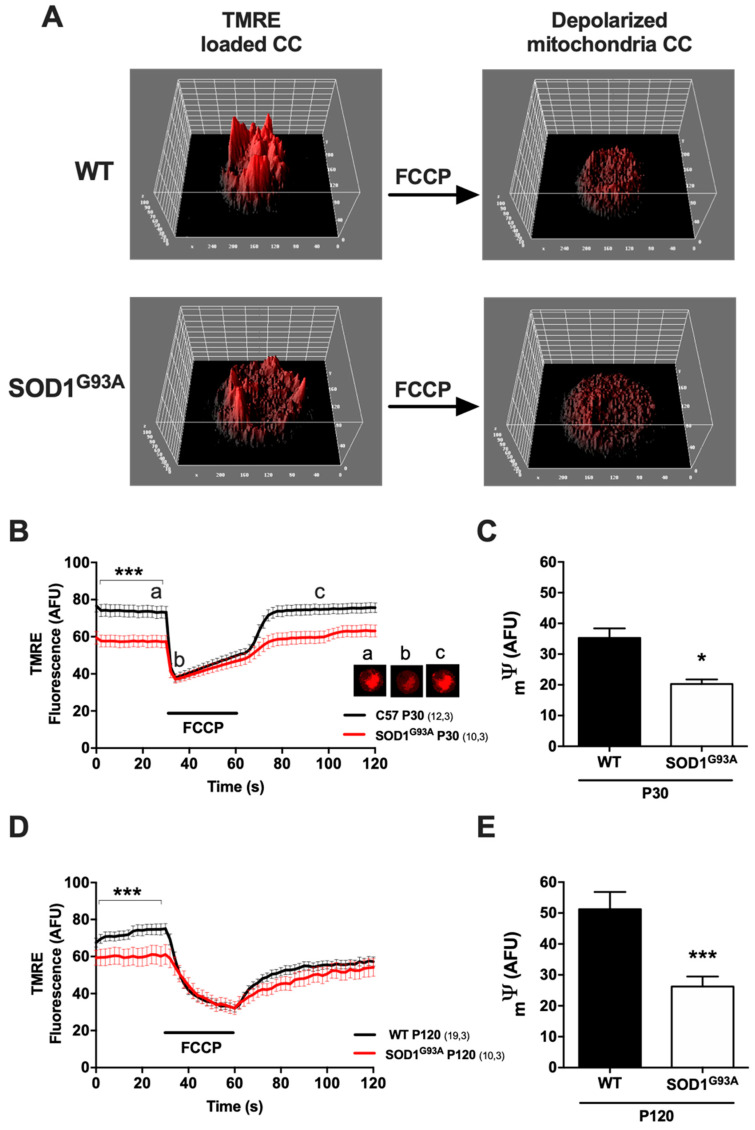
Chromaffin cells from SOD1^G93A^ mice have impaired mitochondrial membrane potential. The _m_ψ was measured in CCs with TMRE. (**A**) A 3D representation of the fluorescence intensity from a single chromaffin cell loaded with TMRE (25 nM). The images on the right show the basal fluorescence at the beginning of the experiments in a CC from WT (**top**) and SOD1^G93A^ (**bottom**) mice. Note that every peak represents the ability of each mitochondrion to load the fluorescent probe based on its _m_ψ. The images on the left show the change in the _m_ψ elicited by the release of the fluorescent probe from the mitochondria induced by the addition of FCCP. (**B**,**D**) The graph represents the average TMRE fluorescence intensity of single cells from P30 (**B**) and P120 (**D**) mice following a protocol of 30 s basal recording, 30s with a FCCP depolarizing pulse and 1 min washout (2 s intervals between data acquisition). The images of the CCs at specific points of the protocol are depicted in **a**–**c**. (**C**,**E**) Bar graphs showing the average variation in _m_ψ in CCs from P30 and P120 mice after depolarization with FCCP. The data represent the mean ± SEM. The number of cells analyzed and cell cultures obtained from different animal are shown in parentheses. Statistical analyses were performed using a Student´s *t* test: * *p* < 0.05; *** *p* < 0.001 vs. WT.

**Figure 6 ijms-22-08194-f006:**
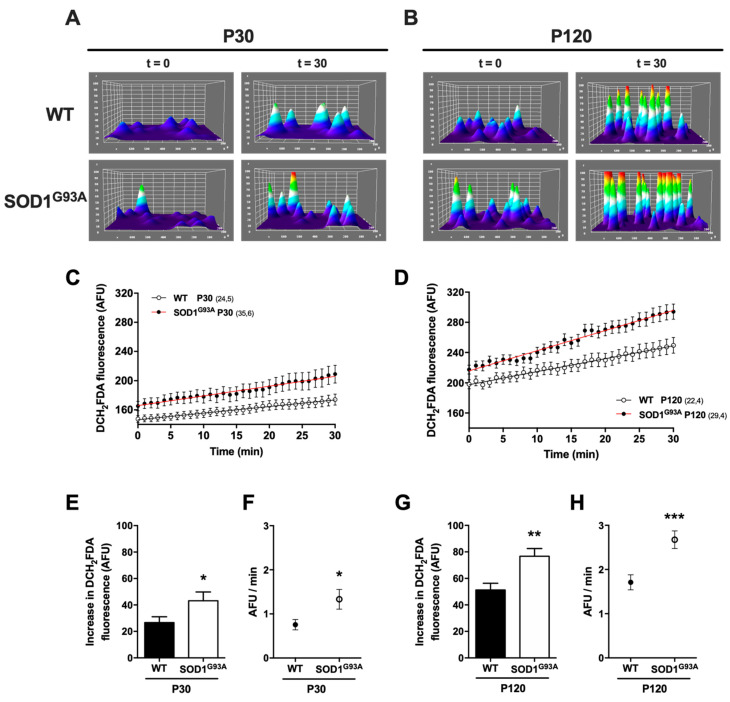
More reactive oxygen species were produced by the chromaffin cells of SOD1^G93A^ mice. The cellular redox balance in CCs was evaluated over 30 min under basal conditions using the DCH_2_FDA fluorescent dye. (**A**,**B**) are 3D reconstructions of single fluorescent dye-loaded CCs at the beginning (t = 0 min, t_0_) and end of the experiment (t = 30 min, t_30_). Each peak represents the fluorescent intensity, which increases continuously over time due to cell activity and is therefore more intense at the end of the experiment (t_30_). (**C**,**D**) represent the average fluorescent intensity in single cells over the 30 min period, acquiring images at 1 min intervals. (**E**,**G**), bar graphs quantifying the increase in fluorescence from t_0_ to t_30_ in cells from P30 and P120 mice, respectively. The average ratio of ROS production, calculated as the slope of the fluorescence increase, is shown in (**F**,**H**). Data represent the mean ± SEM of the number of cells shown in parentheses from four cultures of the same number of animals. The statistical analysis was performed using a Student´s *t* test: * *p* < 0.05; ** *p* < 0.01; *** *p* < 0.001 vs. WT.

**Figure 7 ijms-22-08194-f007:**
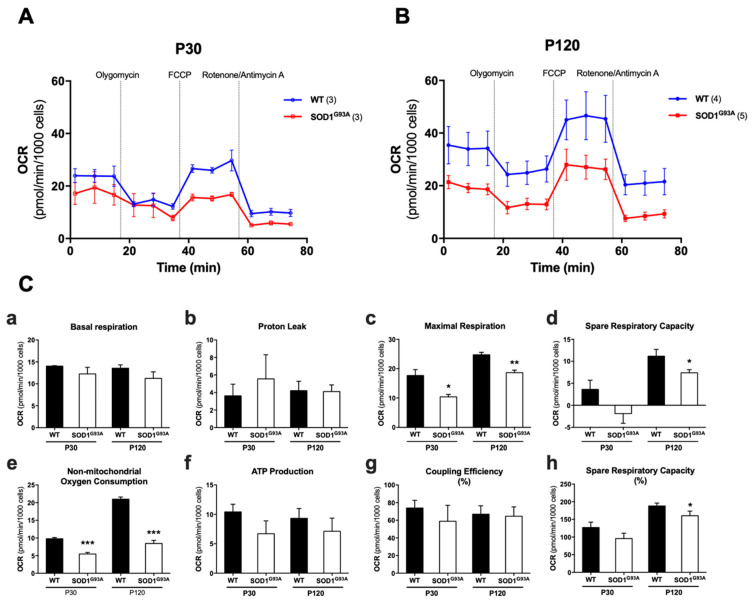
Bioenergetic impairment in chromaffin cells of SOD1^G93A^ mice. The bioenergetic profile of oxygen consumption was analyzed in primary CC cultures from WT (blue line) and SOD1^G93A^ (red line) mice using a seahorse XFp analyzer. The mitochondria stress test was performed on CCs from P30 (**A**) and P120 (**B**) mice, analyzing the oxygen consumption rate (OCR) during the application of drugs targeting the mitochondria: oligomycin, FCCP and rotenone/antimycin A. This procedure allowed us to extract the following parameters in (**C**): (**a**), basal respiration; (**b**), proton leak; (**c**), maximal respiration, (**d**), spare respiratory capacity; (**e**), non-mitochondrial oxygen consumption; (**f**), ATP production; (**g**), percentage coupling efficiency and (**h**); percentage spare respiratory capacity. The bar graphs represent the mean ± SEM of measurements from at least three different cultures, and the statistical analysis was performed using a two-way ANOVA with Šídák’s multiple comparison post hoc test: * *p* < 0.05; ** *p* < 0.01; *** *p* < 0.001 vs. WT at the same age. ^##^
*p* < 0.01 comparing the effect of aging in the WT or SOD1^G93A^ group (panel Cc, Cd, Ce, Ch). These parameters are summarized in the Table 1.

**Figure 8 ijms-22-08194-f008:**
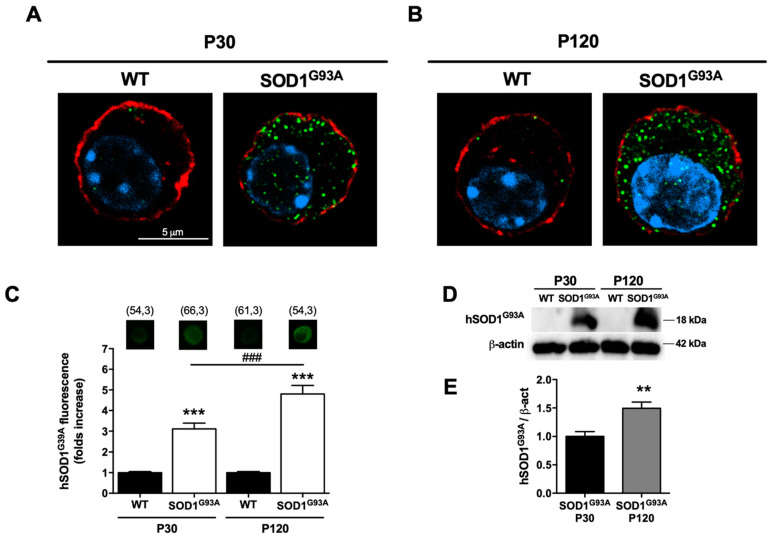
Human SOD1^G93A^ expression in mouse chromaffin cells. Human SOD1^G93A^ expression was explored in CCs by immunofluorescence and in Western blots. (**A**) Confocal images of two isolated CCs obtained from WT and SOD1^G93A^ mice at presymptomatic stages (P30). (**B**) Confocal images obtained at the symptomatic (P120) stage. The cytoskeleton was labeled with Alexa Fluor 568 phalloidin (red), the nucleus with DAPI (blue) and the mutated protein with an anti-human SOD1^G93A^ antibody C4F6 (green). (**C**) Bar graph depicting the mean fluorescence for hSOD1^G93A^ measured in at least three P30 and P120 CC cultures. Data in the graph reflect the epifluorescence intensity, and the insets above each bar show the immunofluorescence from a representative cell: *** *p* < 0.001 vs. WT at same age; ^###^
*p* < 0.001 vs. SOD1^G93A^ P30. (**D**) Immunoblot showing hSOD1^G93A^ expression in adrenal gland medulla, using β-actin as a loading control. (**E**) Quantitative analysis of the mutated protein expression calculated as the hSOD1^G93A^/β-actin ratio: ** *p* < 0.01. The bar graph represents the mean ± SEM of at least five experiments, and statistical analysis was performed using an unpaired Student´s *t* test.

**Figure 9 ijms-22-08194-f009:**
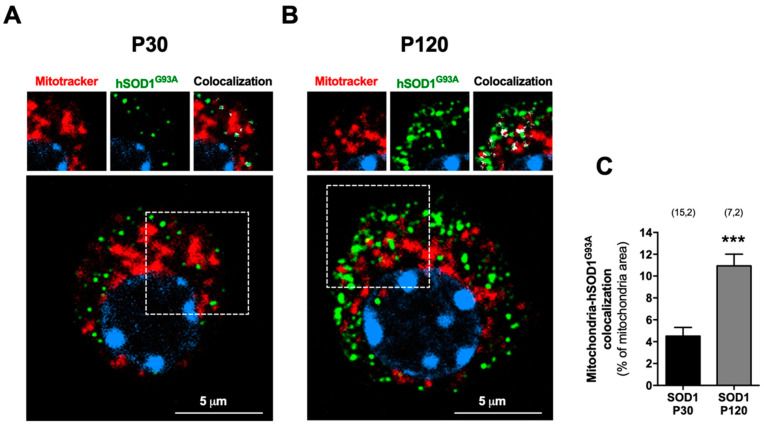
Human SOD1^G93A^-mitochondria colocalization in chromaffin cells during disease instauration: Confocal cross-section of a fixed CC labeled with MitoTracker Red, the anti-human SOD1^G93A^ antibody (green) and DAPI (Blue) at P30 (**A**) and P120 (**B**). Colocalization areas of the red and green signal are seen in yellow. The individual signals within the white inset are shown above the merge image, in which hSOD1^G93A^ and MitoTracker colocalization is highlighted in white. (**C**) The bar graph shows the hSOD1^G93A^-mitochondrial colocalization normalized to the mitochondrial signal: *** *p* < 0.001; Mann–Whitney test. Data are the means ± SEM of the number of cells from two cultures shown in parentheses.

**Figure 10 ijms-22-08194-f010:**
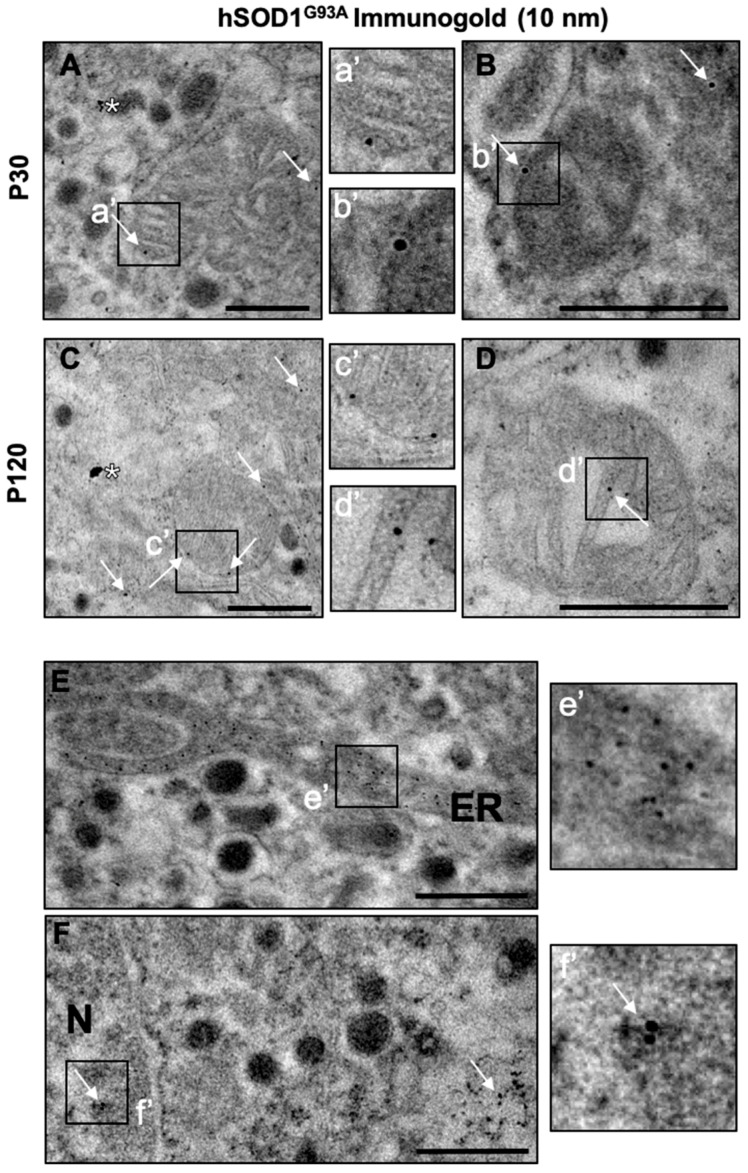
Subcellular localization of the human SOD1^G93A^. TEM images showing the distribution of human SOD1^G93A^ recognized by the C4F6 antibody and a secondary antibody conjugated to 10 nm gold particles. The gold particles were detected in different mitochondrial compartments (intermembrane space, cristae and mitochondrial matrix), both at P30 (**A**,**B**) and P120 (**C**,**D**). The human SOD1^G93A^ protein also accumulates in other organelles, such as (**E**) the smooth endoplasmic reticulum (ER), which shows a large accumulation, and (**F**), the nucleus (N). The human SOD1^G93A^ was also found in the cytosol, lysosomes, peroxisomes and different chromaffin granules (images not shown). Gold particles are indicated by white arrows, while the larger accumulations are signaled with a white asterisk (**A**,**C**). Magnification of the selected areas in each picture are represented as (**a’**–**f’**). Scale bar = 500 nm.

**Figure 11 ijms-22-08194-f011:**
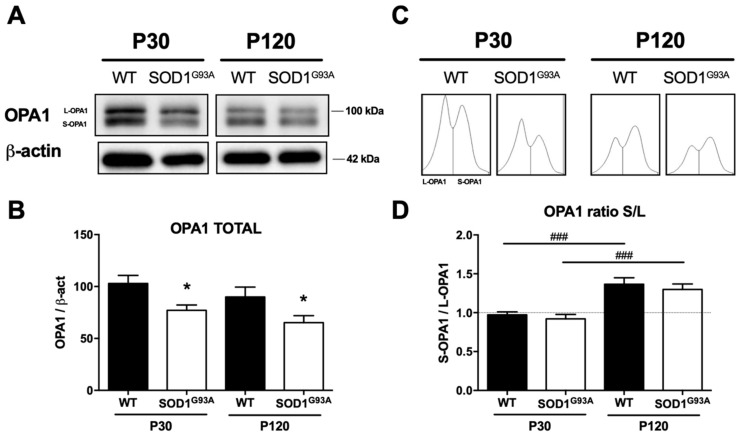
Optic atrophy protein 1 (OPA1) expression is down-regulated in the SOD1^G93A^ mice adrenal medulla from presymptomatic stages, with changes in its cleavage with aging. (**A**) OPA1 expression was analyzed in Western blots of total protein extracted from the adrenal medulla of WT and SOD1^G93A^ mice at P30 and P120. (**B**) Densitometry of the total OPA1 normalized to β-actin (*n* = 4, two-way ANOVA with Šídák’s multiple comparison post hoc test, * *p* < 0.05 relative to WT at same stage). (**C**) Densitometry of the OPA1 bands in A. The first peak of the densitometry represents the long OPA1 isoform (L-OPA1), while the second peak represents the short OPA1 isoform (S-OPA1). (**D**) Bar-chart quantifying the ratio of the OPA1 isoforms ratio measured as the short isoform relative to the long isoform: ^###^
*p* < 0.001 relative to the data at P30.

**Figure 12 ijms-22-08194-f012:**
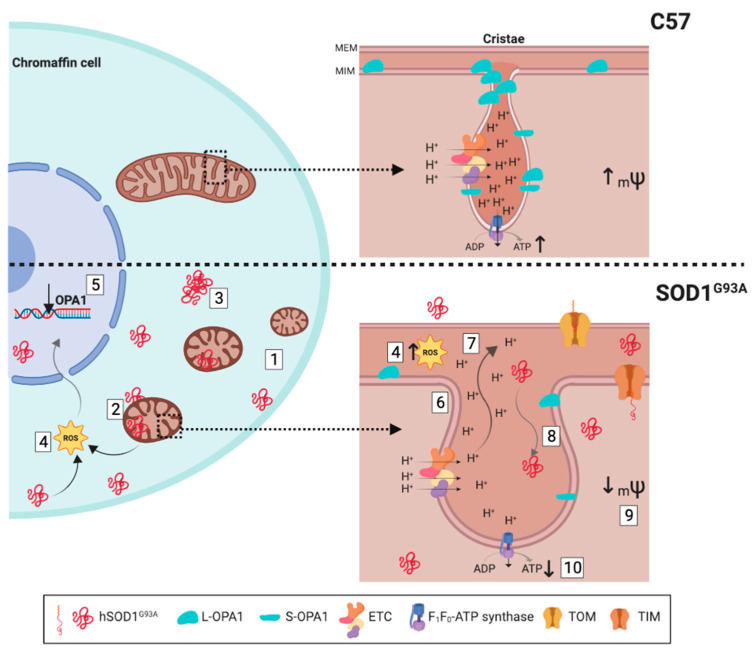
Schematic representation of mitochondrial defects observed in the chromaffin cell of an SOD1^G93A^ mouse. The split-cell drawing represents our observations in the CC of WT mice (**upper part**) and the transgenic SOD1^G93A^ mice (**bottom**). In the cytoplasm of WT CC, there is one healthy mitochondrion with an inset for one cristae magnification (**right**). In the cristae is depicted the location of the OPA1 protein (both S and L isoforms, in blue), the complexes of the electronic transport chain (ETC) and F_1_F_0_-ATP synthase. The optimal management of the cristae junction size allows the proton gradient and the maintenance of _m_ψ. Otherwise, alterations observed in SOD1^G93A^ mice are represented in the bottom part by (1) greater number and smaller size mitochondria; (2) accumulation of the mutated protein hSOD1^G93A^ in the cytoplasm at all mitochondria levels and in the nucleus; (3) formation of hSOD1^G93A^ protein aggregates; (4) increased production of free radicals; (5) decrease of OPA1 gene expression; (6) increase of cristae junctions size; (7) diffusion of cristae components and ions towards the intermembrane space; (8) diffusion of big molecules and proteins as hSOD1^G93A^ inside cristae; (9) loss of _m_ψ with OXPHOS deficiency and (10) decrease in the ATP generation. The deficiencies here present lead to an autonomic nervous system impairment, with inability to overcome stressful events, defects in the bioenergetics profile and inefficient energy consumption. These alterations could be crucial for cell-specific and also patient survival, displaying novel pathological mechanisms and therapeutic targets to overcome ALS.

**Table 1 ijms-22-08194-t001:** Bioenergetic parameters obtained by the Seahorse method.

	P30		P120	
	WT	SOD1^G93A^	WT	SOD1^G93A^
Basal Respiration	14.04 ± 0.05	12.26 ± 0.86	13.55 ± 0.45	11.24 ± 0.86
Proton Leak	3.62 ± 0.76	5.55 ± 1.59	4.22 ± 0.61	4.11 ± 0.43
Maximal Respiration	17.67 ± 1.16	10.39 ± 0.47 *	24.74 ± 0.48	18.63 ± 0.50 **
Spare Respiratory Capacity	3.63 ± 1.20	−1.87 ± 1.27	11.18 ± 0.88	7.39 ± 0.42 *
Non-Mitochondrial Oxygen Consumption	9.78 ± 0.21	5.47 ± 0.25 ***	20.98 ± 0.35	8.46 ± 0.50 ***
ATP Production	10.42 ± 0.75	6.71 ± 1.26	9.34 ± 0.96	7.14 ± 1.28
Coupling Efficiency (%)	73.89 ± 5.02	58.85 ± 10.44	66.85 ± 5.49	64.51 ± 6.16
Spare Respiratory Capacity (%)	127.0 ± 8.81	95.74 ± 8.77	188.0 ± 4.72	160.30 ± 7.60 *

Units are expressed as pmol/min/x1000cells. Data are the mean ± SEM. Two-way ANOVA with Šídák’s multiple comparison post hoc test: * *p* < 0.05, ** *p* < 0.01, *** *p* < 0.001.

## Data Availability

The data that support the findings of this study are available from the corresponding author upon reasonable request.

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
