# Peer review of "Progressive Mitochondrial SOD1G93A Accumulation Causes Severe Structural, Metabolic and Functional Aberrations through OPA1 Down-Regulation in a Mouse Model of Amyotrophic Lateral Sclerosis"

_ijms, 2021, doi:10.3390/ijms22158194_

Round 1

Reviewer 1 Report

In this manuscript, entitled “Progressive mitochondrial SOD1G93A accumulation causes severe structural, metabolic and functional aberrations through OPA1 down-regulation in a mouse model of Amyotrophic Lateral Sclerosis”, Méndez-López et al. identifies OPA1 as a potential target to modulate sympathetic impairment in ALS.

Authors performed experiments in a mouse model of ALS (i.e. SOD1G93A) aiming at demonstrating whether SOD1G93A was accumulated in mitochondria of chromaffin cells, thus distressing mitochondrial fitness and energy metabolism.

I found this manuscript well written and enjoyable. It is also well organized in terms of experimental plan and data presentation. The discussion also offers significant insights and interesting point of view supporting, based on the evidence provided, the so-called “non-autonomous motor neuron death”.

I think that this is a very great manuscript, increasing the current knowledge on ALS pathophysiology and molecular mechanisms and providing novel information exploitable as therapeutic targets for ALS.

I do not have any major points to rise, and I believe that the manuscript is ready to be shared with scientific community and with colleagues in the field. I just have some minor points that authors may found appropriate:

  • In figure 1 – 9, figure 11 and Suppl. Figure 1 and 2, I would change “C57” in “WT” (wild type) and “SOD1” in “SOD1G93A” in both panels and bar graphs.
  • In the discussion authors stated: “It is important to note the need to obtain reductive capacity means that metabolism is inefficient in the mutant mouse, requiring a high glucose, protein and fat consumption to maintain the necessary NADH and FADH supply. This might explain why the SOD1G93A mutant mice lose weight as the disease progresses ( Figure 1). Also, this ADP availability impairment could explain our results of a greater adoption of the orthodox mitochondrial state in relation to the WT mouse”. Maybe authors found of importance to note that acute denervated muscle are related with about the 50% of muscle weight loss (PMID: 30917493) and this has been also found in SOD1G93A mice, in which weight loss correlates with muscular strength deficit (PMID: 21102999).
  • Another important concept is related to the effects of neuronal loss on metabolism of denervated organs. Extensive literature reports these effects in muscle of course (PMID: 17122379; PMID: 17584954; PMID: 34135312). Authors may find of interests correlating previous findings with the advances described in the present manuscript, in terms of mitochondrial metabolism and ROS production in muscle cells and what it has been found in CCs.

I believe that the manuscript is comprehensive, and it can be considered for publication also if authors will decide not to include the previous minor points.

Reviewer 2 Report

The paper by Méndez-López et al. describes the impact of accumulation of the mutated SOD1 G93A on mitochondrial ultrastructure alterations, mitochondrial activity and function in chromaffin cells in ALS murine model.

This is a very interesting paper, I think the approach used to test the mitochondrial activity was well designed. However, a recurrent problem is observed across the paper: the statistical analysis used along the paper should be revised. Indeed, data compared are data from WT and SOD1 mice at 2 time points. Thus ANOVA 2F (or Friedman test) followed by post-hoc test should be more appropriate. In addition paragraph 2.3.6 should be reconsidered (see discussion below).

Below my concerns/suggestions/questions in more details.

Abstract: I would suggest the authors to put more actual data (Values mean ± SD, p values), to sell better the results.

Figure 1: This reviewer appreciates the hard work done on TEM. Measurements/values are given in the text, describing the images shown in Figure 1. Adding histograms showing the distribution of orthodox and condensed mitochondria WT v SOD1 at the 2 time points would be informative (could be stacked histograms). Similarly, another graph could be done for swelling or vacuolisation of the mitochondria (taking into account location and time points, versus WT/SOD1), mitochondrial sprouting etc. Appropriate statistical analysis could be then performed on these data (ANOVA 2F + Post-hoc test), thus emphasising the results.

Figure 2: Graph and statistical analysis: Not clear whether one image was considered as n=1. It needs to be clarified. And if it is the case, then comparison should be per animal, not per images nor per cells.

Figure 3: Again need to clarify how the statistical analysis has been done (taking the average of the 3 mice or comparing images or mitochondria). As there are 2 groups of animals at 2 time points, an 2F-ANOVA (or corresponding non-parametric test if not enough n) followed by post hoc test should be done. Simple t.test comparison is not adapted. In Fig 3D brackets showing the number of cristea are missing.

Figure 4: Here too, 2F-ANOVA (or corresponding non-parametric test if not enough n) followed by post hoc test should be done. Mann-Whitney t test not appropriated.

Figure 6: ROS scavenger should be used here as a negative control for DCFHDA probe. To link the ROS production with Mitochondrial dysfunction, it would be interesting to see if ROS production is affected differently on SOD1 and WT cells when adding FCCP, ADP, Oligomycin A, antimycin A.

Figure 5, Table 1 and Figure 7: This is a very good investigation done here (Again, check the appropriate stats). Question: what happen it terms of Mitochondrial respiration/activity and/or ROS production when mutated SOD1 aggregates are added to the healthy mitochondria?

Figure 8: To explore whether “SOD1G93A is a possible cause of mitochondrial damage”, an inducible SOD1G93A expression cell model would have helped to finalise this question. Otherwise, the observations done here are more a correlation than causative.

Figure 11: Again check the statistical analysis. Fig11C, not sure it brings something to the paper.

Paragraph 2.3.6. Fewer ubiquitinylated proteins in chromaffin cells from the adrenal medulla of SOD1G93A mice, in the absence of changes in the proapoptotic bax protein

The authors tried to wrap up the story, opening on the downstream effect of the mitochondriopathy on autophagy/cell death. This part is less complete than the rest of the paper. As the whole topic was around mitochondria, investigating specifically markers of mitophagy would have been more relevant (Pink etc). The authors could reinforce this paragraph by looking at mitophagy feature on electron microscopy: Was there an accumulation of degrading/mitophagy features observed by electron microscopy in SOD1 CC cells?

Then cell death is briefly approached with bax investigation. Is there an actual loss of CC in SOD1 mice? What percentage? This section should be completed following up other death markers: apoptosis with caspases 3/9, Bcl2, Cyt C, necrosis (cleaved PARP1 etc). This may even go out of the topic of the current paper. Considering that the current paper is already quite dense in data, may be the authors should consider finalising on mitochondrial topic, instead on opening other topics.

Material and methods, Statistical analysis:

Line 966-967: confusing sentence: “The data with a normal distribution were compared using a non-parametric Student t test,…”

Not sure it is correct to use one image or one cell as a n=1,  it should be n=1 animal.

Discussion: Should be shortened (7 pages for discussion, need to reduce by 2/3 – ½), and sometimes a bit too speculative (eg. Line 562-566: “it is logical to think that if the cells have a low basal….)

Discussion line 572-573: “This is particularly evident when the cells of the mutant SOD1G93A mice are subjected to an even greater collapse of the my in the presence of FCCP.” Looking at figure 5B,D, I don’t think it is correct to say the collapse of my is greater in SOD1 cells: the deltaPsi is at a lower basal level to start with, then uncoupling the mitochondria with FCCP make the deltaPsi drop at the same level as WT mitochondria treated with FCCP. But I agree that the mitochondrial respiration is less efficient in SOD1 cells.

Line 981 and line 988: symbol problem

Figure 12:  Need to show/symbolize in the zoomed square the decrease in ATP production in SOD1, as well as the increase in ROS production.

Bibliography: could cite more recent review/original papers on ALS multisystemic disease.

Minor:

Introduction: line 50: “becoming the best studied mouse model”. I guess the authors mean the “most studied”.

Line 109: “As SOD1 is a ubiquitous enzyme” should be” As SOD1 is an ubiquitous enzyme”
